



# Impacts of Inter-basin Water Diversion Projects on the Feedback Loops of Water Supply-Hydropower Generation-Environment Conservation Nexus

Jiaoyang Wang[1], Dedi Liu[1,2,3], Shenglian Guo[1], Lihua Xiong [1], Pan Liu [1], Hua Chen [1], Jie Chen [1], Jiabo Yin [1,2], and Yuling Zhang[1]

[1] State Key Laboratory of Water Resources Engineering and Management, Wuhan University, Wuhan 430072, China

[2] Hubei Provincial Key Lab of Water System Science for Sponge City Construction, Wuhan University, Wuhan, 430072, China

[3] Department of Earth Science, University of the Western Cape, Bellville, Republic of South Africa.

*Correspondence: Dedi Liu (dediliu@whu.edu.cn)*

**Abstract.** To balance water resource distribution among different areas, inter-basin water diversion projects (IWDPs) have been constructed around world. The unclear feedback loops of water supply-hydropower generation-environmental conservation (SHE) nexus with IWDPs increase the uncertainty in the rational scheduling of water resources for the water receiving and water donation areas. To address the different impacts of IWDPs on the dynamic SHE nexus and explore collaborative states, a framework was proposed to identify these impacts across the multiple temporal and spatial scales in a reservoirs group. The proposed approach was applied to the Hanjiang River Basin (HRB) in China as a case study. Multiple temporal and spatial scales runoffs from HRB were provided through the Variable Infiltration Capacity hydrological (VIC) model. And multi-level ecological flows and their corresponding multi-level ecological flow standards were also determined by the Modified Tennant Method Based on Multilevel Habitat Conditions (MTMMHC) method. 30 scenarios were set and modeled in a multisource input-output reservoir generalization model. Differences between scenarios were quantified with a response ratio indicator. The results indicated that: there are negative feedbacks between water supply (S) and hydropower generation (H), between S and environmental conservation (E) while positive feedbacks between H and E without IWDPs. The negative feedbacks of S on H and the positive feedbacks of E on H are weakened or even broken in abundant water periods. Water donation has negative impacts on feedback loops, while water receiving has positive impacts on these feedbacks. Feedback loops exhibit intrinsic similarity and stability across different time scales. Feedbacks in reservoirs with regulation function remain stable under the varying inflow conditions and feedbacks for downstream reservoirs are influenced by their upstream reservoirs, especially in low flow periods. The proposed approach can help quantify the impacts of IWDPs on SHE nexus and contribute to the sustainable development of SHE nexus.

## 1 Introduction

Water resources are fundamental to life, as well as economic and social development (MacGREGOR, 1963). Water supply, hydropower generation, and environmental conservation constitute the three primary components of water resource utilization in a basin (Chung et al., 2021), delivering substantial economic, social, and ecological benefits to both humanity and nature. However, over the past 70 years, global water resources have been rapidly consumed and utilized, due to the increasing human demand and climate change, leading to complex supply-demand conflicts (Tauro, 2021; Wang et al., 2024). Water supply, hydropower generation, and environmental conservation compete, coordinate, and are interdependent with each other, and intricate relationships can be found among them (Stickler et al., 2013). The interdependencies among these water supply (S),



hydropower generation (H), and environmental conservation (E) components are referred as an SHE nexus (Endo et al., 2017; FAO., 2014; Sanders and Webber, 2012). Identifying the SHE nexus can elucidate the trajectory of water resources system evolution under various water resource management strategies, balance the relationships among water users, and promote sustainable resource use and ecological health (Mansour et al., 2024; Zhao et al., 2021).

The current studies on the nexus primarily focus on the three fundamental resources: water, energy, and food (Conway et al., 2015; Quer et al., 2024; Wang et al., 2023). The SHE nexus refines the water-energy-food nexus and emphasizes the basin-scale water resource management (Chen et al., 2020). Most of the studies on SHE nexus take reservoirs as nodes, and primarily focus on multi-objective optimization of basin-wide water resource scheduling (Khalkhali et al., 2018; Qiu et al., 2021; Tang et al., 2024). Through game-theoretical analyses among components, they aim to identify feedback between their paired
components. From the perspective of reservoir nodes under scrutiny, current research primarily focuses on single reservoirs (Wu et al., 2021), virtual reservoirs (Chen et al., 2020), and cases of two connected reservoirs (Khalkhali et al., 2018) and few of them concern on the reservoirs group with different priority functions. The different priority functions of reservoirs lead to the different SHE nexus. It is conducive to deciphering the nexus of and the directional changes within the SHE system, that the reservoirs are located in different locations within a basin, prioritizing different objective functions. Moreover,
quantification of E component often relies on the Tennant method (Tennant, 1976; Tharme, 2003) to estimate ecological flows while neglects the temporal and spatial variations. some of the E components only contain urban and rural ecological water use, and neglects the in-stream ecological flows (Chen et al., 2020). There is often not a straightforward positive or negative correlation between water supply, hydropower generation, and environmental conservation components (Zitzler, 2007). The feedback loops among components in a system are not static but changes or breakthroughs from different time-space
perspectives (Keyhanpour et al., 2021). The components S, H, and E interact dynamically over time and space (Dong et al., 2019), inevitably leading to changes in the feedback loops of SHE nexus. However, studies on these changes in the SHE nexus are relatively scarce. Identifying collaboration within competitive loops or competition within collaborative loops across various time-space scales enhances understanding of the dynamic changes in the SHE nexus. And it also provides strategies for dealing with competition among different users in actual water management. Therefore, it is critical to investigate the
bidirectional and dynamic feedback loops of the SHE nexus across multiple temporal and spatial scales.

Due to frequent extreme events and intensive human activities, the spatial and temporal distribution of water resources exhibit more and more unevenness (Wang et al., 2024). Imbalance of water supply-demand has widely spread all over the world at any time. Inter-basin water diversion projects (IWDPs) have been widely implemented to solve the imbalance (Siddik et al., 2023) through transferring water resources from water-rich areas (i.e., water donating area) to water-deficient regions
(i.e., water receiving area) through channels and other hydraulic engineering works. The initiatives of the IWDPs seek to alleviate the imbalance among different basins but also result in notable changes of the water resource systems in both the source and receiving areas (Long et al., 2020). Many studies have extensively examined the receiving effects of IWDPs on the three components (Tang et al., 2022; Tao et al., 2008; Wei et al., 2024), as well as on the comprehensive evaluation of water resource systems (Kattel et al., 2019; Zhao et al., 2017) and multi-factor risk assessment of water donating areas (Bai et al.,
2023; Mu et al., 2024; Yang et al., 2023) at different time and space scales. It was found that the dynamic planning and operation of IWDPs exert significant external impacts on the SHE system, inevitably leading to the system's "change-response-reconstitute" process. These impacts changed the feedback loops among components of the SHE system. Additionally, studies have primarily emphasized single water donating or receiving impacts, overlooking the different impacts of IWDPs on the SHE nexus and the comprehensive effects of multi-IWDPs. Water management regulations with IWDPs has been becoming
one of the focuses in the SHE nexus (Mok et al., 2015). The current studies on this issue have primarily examined the optimal water allocation methods for negotiations among water users in donating and receiving areas. They often employ case study





approaches (e.g., interviews, field studies, policy reviews, and surveys) (Zhao et al., 2017) or inter-basin water resource allocation models (Ouyang et al., 2020; Wu et al., 2022). However, most of these studies have still oversimplified the interactions among these three components as only competitive (Yan et al., 2020). Finding the changes on the feedback loops

with IWDPs and collaborations following the feedback loop changes are crucial steps in improving water dispatching and management in both donating and receiving areas.

One of the aims in this study is to identify the different impacts of IWDPs across multiple temporal and spatial scales on the dynamic SHE nexus in reservoirs group with different priority functions. And another is to explore a way to search collaborative states in the feedback loops of SHE nexus. The research framework and methods are presented in Section 2, and

our case study to verify the proposed framework are detailed in Section 3. Section 4 covers the results and discussion, and conclusions are drawn in Section 5.

## 2 Methodology

### 2.1 Research framework

To address the impacts of IWDPs across the multiple temporal and spatial scales on the dynamic SHE nexus, multiple temporal

and spatial scales runoffs from the water donating basins are provided through a distributed hydrological model. And multi-level ecological flows and their corresponding multi-level ecological flow standards are also determined according to an available method with spatial-temporal variability. To facilitate the identification of the impacts of IWDPs on SHE nexus, scenario experiments are set by "with/without IWDPs". In order to take the different clusters of IWDPs into account, scenario experiments are classified by the impacts of IWDPs on water donation area, on water receiving area or on an area with both

water donation and water receiving if there are IWDPs. To evaluate the feedback loops of the SHE nexus, the priority order of S, H, and E are iteratively set in all reservoir nodes. We set different types of the highest priority in S, H, and E (i.e., S-Priority, H-Priority, and E-Priority) and take the standard scheduling rules as reference scenarios. All scenarios are modeled in a multisource input-output reservoir generalization model, and differences between scenarios are quantified with a response ratio indicator. And the feedback loops with the different impacts of IWDPs are identified through a response ratio indicator. To

explore the collaborative states, positive mutation in a response ratio across time-space is found between pairwise components of SHE. Thus, our research framework is illustrated as Figure 1.



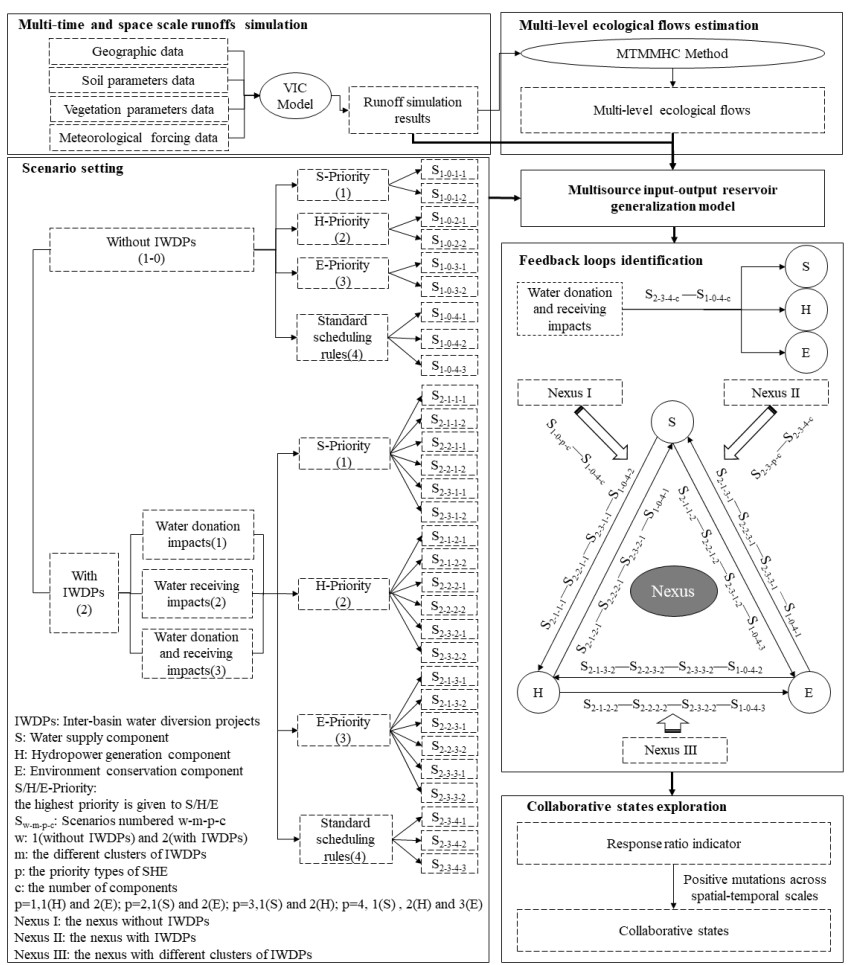

**Figure 1. Framework to identify the impacts of different IWDPs on the feedback loops of SHE nexus.**

### 2.2 The Variable Infiltration Capacity hydrological model

To simulate runoff results at multiple temporal and spatial scales , the Variable Infiltration Capacity (VIC) hydrological model
is selected. VIC model is a large-scale distributed hydrological model based on the spatial distribution grid of Soil Vegetation
Atmospheric Transfer Schemes (SVATS) (Liang, et al., 1994). It has been widely application in runoff simulations across
various basins worldwide, consistently yielding outstanding results. It excelled at simulating both the energy balance and water
balance between the land and atmosphere, thereby addressing the oversight of energy processes in traditional hydrological

models. There are five steps to construct a VIC model (Koohi et al., 2022): ① collect and organize data; ② preprocesses of the
VIC model; ③ construct VIC model of the selected basin; ④ run the catchment module; ⑤ parameter calibration and validation.
During the calibration process, important parameters highlighted in Table 1 are automatically calibrated using MATLAB to
achieve the optimal parameter combination.

**Table 1**. Characteristics of parameters for model optimization (Gou et al., 2020).

| No. | Parameter | Brief description | Unit | Range |
|-----|-----------|-------------------|------|-------|
| 1 | $B$ | The power of the equation for the variable infiltration curve | / | [0,0.4] |





| 2 | $D_{smax}$ | The maximum baseflow velocity | mm/day | [0,30] |
|---|---|---|---|---|
| 3 | $D_s$ | The ratio of the nonlinear baseflow to $D_{smax}$ | / | [0,1] |
| 4 | $W_s$ | The ratio of nonlinear baseflow to saturated soil moisture content when it occurs | / | [0,1] |
| 5 | $d_1$ | Thickness of the top layer of soil | m | [0.05,0.1] |
| 6 | $d_2$ | Thickness of the second layer of soil | m | [0,2] |
| 7 | $d_3$ | Thickness of the third layer of soil | m | [0,2] |


In order to verify the accuracy of the runoff simulation results, the simulations need to be compared with the observations. Three widely used quantitative indices of numerical differences are selected, and they are the Nash-Sutcliffe efficiency coefficient ($NSE$, Nash and Sutcliffe, 1970), Coefficient of determination ($R^2$, Rousseeuw and Leroy, 1987), and Percent bias ($PBLAS$, Bland and Altman, 1986):


$$NSE = 1 - \frac{\sum_{t=1}^{T}\left(Q_t^o - Q_t^s\right)^2}{\sum_{t=1}^{T}\left(Q_t^o - \overline{Q^o}\right)^2} \qquad (1)$$

$$R^2 = \frac{\left[\sum_{t=1}^{T}\left(Q_t^o - \overline{Q^o}\right)\left(Q_t^s - \overline{Q^s}\right)\right]^2}{\sum_{t=1}^{T}\left(Q_t^o - \overline{Q^o}\right)^2 \sum_{t=1}^{T}\left(Q_t^s - \overline{Q^s}\right)^2} \qquad (2)$$

$$PBIAS = \frac{\sum_{t=1}^{T}\left(Q_t^o - Q_t^s\right)\times 100}{\sum_{t=1}^{T} Q_t^o} \qquad (3)$$

where, $Q_t^o$ and $Q_t^s$ are the observed and simulated runoff results at $t$th month, m³/s. $\overline{Q^o}$ and $\overline{Q^s}$ are the average of the observed and simulated runoff results over the whole period $T$, m³/s. $NSE \in (-\infty, 1]$, the closer $NSE$ is to 1, the better the simulations are.

The $NSE$ of the simulations greater than 0.5 is acceptable. $R^2 \in [0,1]$, $R^2$ approaching 1 meant the simulations are equal to the observations. $PBLAS$ is utilized to quantify the cumulative deviation between the simulations and observations. $PBLAS$ lager than 0 meant that the simulations are generally small, and vice versa, the simulations are generally large. When $|PBLAS| < 25\%$, the runoff simulation results are acceptable.

After getting the acceptable runoff simulation results at the selected hydrological stations, the dam discharge from the
primary reservoir and the interval runoff of each pair reservoirs are estimated according to the catchment area ratio of each reservoir with its upstream and downstream hydrological stations. The calculation formulas are as follows:

$$Q_{i,t} = \begin{cases} \dfrac{Q_{d,i,t}^s \times A_i}{A_{d,i}}, i = 1 \\[4mm] Q_{u,i,t}^s + \dfrac{\left(Q_{d,i,t}^s - Q_{u,i,t}^s\right)\times\left(A_i - A_{u,i}\right)}{\left(A_{d,i} - A_i\right)}, i > 1 \end{cases} \qquad (4)$$

$$\Delta Q_{i,t} = Q_{i,t} - Q_{i-1,t}, i > 1 \qquad (5)$$

where $Q_{i,t}$ is the dam discharge form the $i$th reservoir at $t$th period, m³/s; $Q_{u,i,t}^s$ and $Q_{d,i,t}^s$ are the runoff simulation results of



the upstream and downstream hydrological stations of the $i$th reservoir at $t$th period, m$^3$/s; $A_i$ is the catchment area of $i$th

reservoir, m²; $A_{u,i}$ and $A_{d,i}$ are the catchment areas of the upstream and downstream hydrological stations, m².

### 2.3 The Modified Tennant Method Based on Multilevel Habitat Conditions method

In order to establish a multi-level ecological flow standard to aid in evaluating river ecological health, the multi-level ecological

flows are estimate by the MTMMHC method. The MTMMHC method (Li and Kang, 2014) modifies the Tennant method based

on three parameters: average periodic flow, water period, and percentage. Indeed, the MTMMHC method can avoid the impacts

of extreme inter-annual flow events and uneven intra-annual distribution. This enables the calculation of different guarantee

rates for various river sections, water years (e.g., wet, normal, and dry years), and months. It reflects the temporal and spatial

variability of ecological flows, and provides a comprehensive and reasonable multi-level ecological flows standards. The steps

of the MTMMHC method are as follows.

①  The year groups are divided into wet years (P<25 %), normal years (25 %≤P≤75 %), and dry years(P>75 %) firstly.

Then, a flow duration curve (FDC, Franchini et al., 2011) is constructed using the total-period method based on daily average

flows. Finally, the average of flows corresponding to the 90th and 95th percentiles of the FDC ($Q_{(90)xy}$ and $Q_{(95)xy}$, m$^3$/s) for the

$y$th month of the $x$th year is taken as the Minimum Ecological Flow ($MEF_{xy}$, m$^3$/s). The formula is as follows:

$$MEF_{xy} = \frac{Q_{(90)xy} + Q_{(95)xy}}{2} \tag{6}$$

②  The MTMMHC method takes 50 % flow of the FDC ($Q_{(50)xy}$, m$^3$/s) for the $y$th month of the $x$th year as the maximum

of the Optimum Ecological Flow ($OEF_{xy\,(max)}$, m$^3$/s). According to the Tennant method, the ecological flows are assumed to be

ten levels, and the minimum of the Optimum Ecological Flow ($OEF_{xy\,(min)}$, m$^3$/s) is set as the level six, and the formulas are as

follows:

$$OEF_{xy(max)} = Q_{(50)xy} \tag{7}$$

$$OEF_{xy(min)} = \frac{5Q_{(50)xy} + 4MEF_{xy}}{9} \tag{8}$$

③  The MTMMHC method computes ecological flows at all levels using the arithmetic difference between $MEF_{xy}$ and

$OEF_{xy\,(min)}$. The MTMMHC method eliminates the classification of $OEF_{xy\,(min)}$—$OEF_{xy\,(max)}$, resulting in the grading number of

ecological flows to be $R+1$. The mode of all the grading number of selected stations is taken as the grading number $R$:

$$R = \text{Mode}\left(\text{Average}\left(m_{xy}\right)\right) \tag{9}$$

$$m_{xy} = \text{Round}\left(\frac{5}{9} \times \frac{Q_{(50)xy} - MEF_{xy}}{0.1 \times Q_{(50)xy}}\right) + 1 \tag{10}$$

where, $m_{xy}$ is the grading number between $MEF_{xy}$ and $OEF_{xy(min)}$ in the $y$th month and $x$th year; Mode($\cdot$) , Average($\cdot$) , and  Round($\cdot$)

are the functions which return the most frequently occurred number in Average ($m_{xy}$), the average of $m_{xy}$, and the nearest integer.

④  Based on the hierarchical idea of arithmetic progression, a range of EF criteria can be defined as follows:

$$EF_{xy(r)} = MEF_{xy} + \frac{5}{9} \times \frac{r-1}{R-1} \pi \left(Q_{(50)xy} - MEF_{xy}\right) \tag{11}$$

where, $EF_{xy(r)}$ is the $r$th level ecological flow in the $y$th month of the $x$th year, m$^3$/s.





### 2.4 The Log Response Ratio method for identifying feedback loops

#### 2.4.1 Water supply, hydropower generation and environment conservation indexes

To evaluate the state of S, H, and E, the water supply volume, hydropower generation, and ecological flow satisfaction rate as indexes of the three components are set. The formulas are as follows.

① Regional water supply volume:

$$V_{s,i,t} = Q_{s,i,t} \times \Delta t = V_{i,\,t} - V_{i,\,t+1} + \left(Q_{\text{out},i-1,t} + \Delta Q_{i,t} + Q_{\text{re},i,t} - Q_{\text{out},i,t} - Q_{\text{do},i,t}\right)\Delta t - I_{i,t} \tag{12}$$

where, $V_{s,i,t}$ is the regional water supply volume, m³; $Q_{s,i,t}$ is the regional water supply flow, m³/s; $\Delta t$ is the time interval, s; $V_{i,t}$ and $V_{i,t+1}$ are the storage of the $i$th reservoir in period $t$ and $t+1$, m³; $Q_{\text{out},i-1,t}$ is the water release from the ($i$-1)th reservoir in period $t$, m³/s; $\Delta Q_{i,t}$ is the flow of the intervening basin between the ($i$–1) th and $i$th reservoirs in period $t$, m³/s. $Q_{\text{re},i,t}$ is the

water receiving from IWDPs, m³/s, and $Q_{\text{do},i,t}$ is the water donation for IWDPs, m³/s. $I_{i,t}$ is the sum of evaporation and seepage losses from the reservoir in period $t$, m³, respectively.

    ② Hydropower generation:

$$E_{i,t} = \sum_{t=1}^{T} N_{i,t}\Delta t \qquad N_{i,t} = K_i Q_{e,i,t} H_{i,t} \tag{13}$$

where, $E_{i,t}$ is the hydropower generation of the $i$th reservoir, kW·h; $N_{i,t}$ is the output of the $i$ th reservoir in the $t$ th period,

kW; $K_i$ is the hydropower generation efficiency of the $i$th reservoir; $Q_{e,i,t}$ and $H_{i,t}$ are the release discharge for hydropower generation, m³/s, and the average hydropower head of the $i$th reservoir in period $t$, m, respectively.

    ③ Ecological flow satisfaction rate is used to evaluate the satisfaction of intra-river flow to multi-level ecological flow standard. It is quantified through the segmented linear affiliation function:

$$EFSR_{xy} = \begin{cases} 0 & EF_{xy} \leq \dfrac{E_{xy(1)}}{2} \\[2ex] \dfrac{1}{R+1}\left(\dfrac{EF_{xy} - \dfrac{E_{xy(1)}}{2}}{E_{xy(1)} - \dfrac{E_{xy(1)}}{2}}\right) & \dfrac{E_{xy(1)}}{2} < EF_{xy} \leq E_{xy(1)} \\[2ex] \dfrac{1}{R+1} + \dfrac{1}{R+1}\left(\dfrac{EF_{xy} - E_{xy(1)}}{E_{xy(2)} - E_{xy(1)}}\right) & E_{xy(1)} < EF_{xy} \leq E_{xy(2)} \\[2ex] \dfrac{2}{R+1} + \dfrac{1}{R+1}\left(\dfrac{EF_{xy} - E_{xy(2)}}{E_{xy(3)} - E_{xy(1)}}\right) & E_{xy(2)} < EF_{xy} \leq E_{xy(3)} \\[2ex] \cdots \qquad\qquad\qquad \cdots \\[1ex] \dfrac{R-1}{R+1} + \dfrac{1}{R+1}\left(\dfrac{EF_{xy} - E_{xy(R-1)}}{E_{xy(R)} - E_{xy(R-1)}}\right) & E_{xy(R-1)} < EF_{xy} \leq E_{xy(R)} \\[2ex] \dfrac{R}{R+1} + \dfrac{1}{R}\left(\dfrac{EF_{xy} - E_{xy(R-1)}}{E_{xy(R)} - E_{xy(R-1)}}\right) & E_{xy(R)} < EF_{xy} \leq E_{xy(R+1)} \\[2ex] 1 & E_{xy(R+1)} < EF_{xy} \end{cases} \tag{14}$$





where, $EFSR_{xy} \in [0,1]$, is the ecological flow satisfaction rate in the $y$th month of the $x$th year. $E_{xy(1)}$, $E_{xy(R)}$ and $E_{xy(R+1)}$ are

$MEF_{xy}$, $OEF_{xy(min)}$ and $OEF_{xy(max)}$, respectively.

### 2.4.2 The Multisource Input-Output Reservoir Generalization (MIORG) model for a reservoirs group

Reservoirs can determine S, H, and E according to their scheduling rules. To quantify the differences of indexes with different impacts of IWDPs in reservoir nodes, MIORG models for reservoirs group are developed. For a single reservoir, the inputs

generally refer to the inflow from the upstream and water receiving from IWDPs. The outputs from this MIORG model refer to regional water supply (i.e., domestic water supply, production water supply and ecological water supply for the outside of the river), water donation for IWDPs, evaporation and seepage losses, water release from the reservoir. The multisource input-output to a single reservoir is shown in Figure 2.

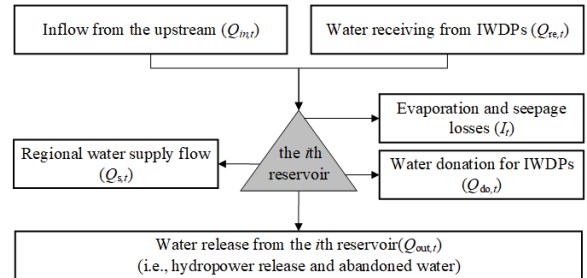

**Figure 2. The multisource input-output to a single reservoir.**

According to the principle of water balance, the MIORG model for a single reservoir is developed as follows:

$$V_{t+1} = V_t + \left( Q_{in,t} + Q_{re,t} - Q_{s,t} - Q_{out,t} - Q_{do,t} \right) \Delta t - I_t \qquad (15)$$

For a reservoirs group, the inputs to $i$th reservoir can be categorized into: water release from the upstream reservoir (i.e., the ($i$-1) th reservoir), the flow of the intervening basin and water receiving from IWDPs. And the outputs from $i$th reservoir in

a reservoirs group are same as those from a single reservoir. The multisource input-output to $i$th reservoir in a reservoirs group is shown in Figure 3. The MIORG model for the $i$th reservoir in a reservoirs group is:

$$V_{i,t+1} = V_{i,t} + \left( Q_{out,i-1,t} + \Delta Q_{i,t} + Q_{re,i,t} - Q_{s,i,t} - Q_{out,i,t} - Q_{do,i,t} \right) \Delta t - I_{i,t} \qquad (16)$$





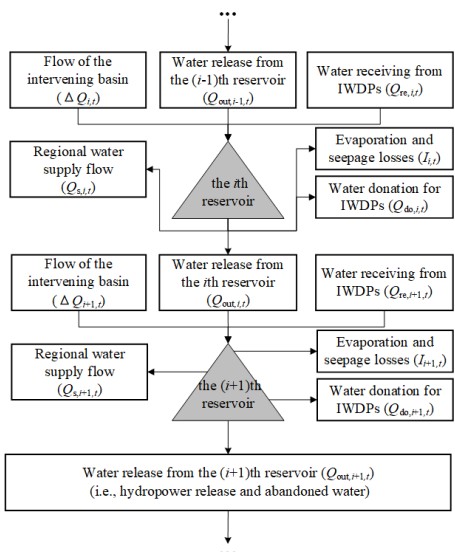

**Figure 3. The multisource input-output to reservoirs in a reservoirs group.**

### 2.4.3 The Log Response Ratio method

To analyse the feedback loops in Nexus I, Nexus II and Nexus III in Figure 1, the log response ratio (*LRR*) quantization method (Patrick et al., 2022) is used to quantify the responses of S, H, and E with different clusters of IWDPs. The formula is as follows:

$$LRR_n = \ln\left(\frac{\left(r_{c(n)} - r_n\right) + r_n}{r_n}\right) = \ln\left(\frac{r_{c(n)}}{r_n}\right) \tag{17}$$

where $LRR_n$ is the log response ratio of the $n$th component. $r_n$ is the value of the $n$th index, and $r_{c(n)}$ is the value of the $n$th index need to be compared. $r_{c(n)}$ and $r_n$ are both greater than or equal to zero. $LRR_n>0$ indicated that the $n$th component is optimized, and the larger the $LRR_n$, the more positive changes in the $n$th component.

### 2.5 Scenario setting

To identify the impacts of different clusters of IWDPs on the SHE nexus, scenarios are set according to the following three aspects: with or without IWDPs (i.e., two types for IWDPs), different clusters of IWDPs (i.e., four clusters for the above two types), and the priority orders of S, H, and E. As there are three components for the highest priority, six scenarios can be obtained through the combination of the three components. As all S, H, and E are determined from standard scheduling rules, there are also three types for the standard scheduling rules. Combined with the types of different clusters of IWDPs, there will be a total of 30 scenarios (i.e., 4 clusters of IWDPs × 6 types for the highest priority combinations +2 types for IWDPs × 3 types for standard scheduling rules) as listed in Table 2. Specifically, to iteratively set the priority orders of S, H, and E, all three components are all in standard scheduling rules firstly. Secondly, the highest priority is set to water supply (as denoted by S-Priority), with the regional water supply increased to 120 %. And thirdly, hydropower generation (H-Priority) is prioritized to achieve the maximum output during the planned period. Finally, environmental conservation (E-Priority) is addressed through ensuring that the reservoir outflow meets $OEF_{xy(max)}$.





To analyse the feedback loops of SHE nexus without IWDPs, the differences between the $S_{1-0-p-c}$ and $S_{1-0-4-c}$ scenarios are
determined (i.e., the feedback loops of Nexus I as shown in Figure 1.). To analyse the feedback loops with IWDPs (i.e., the
feedback loops of Nexus II as shown in Figure 1.), the differences between the $S_{2-3-p-c}$ and $S_{2-3-4-c}$ scenarios are determined.
Thus, the differences between Nexus I and Nexus II can figure out the impacts of IWDPs on the SHE nexus. To identify the
SHE nexus with different clusters of IWDPs (i.e., the feedback loops of Nexus III as shown in Figure 1.), the differences
between $S_{2-m-p-c}$ and $S_{1-0-4-c}$ scenarios are determined. Thus, the differences between Nexus I and Nexus III can figure out the
impacts of IWDPs on the SHE nexus. $S_{1-0-4-c}$ and $S_{2-3-4-c}$, are the baseline scenarios for distinguishing Nexus I, Nexus III, and
Nexus II. In the same way, to clarify the impacts of IWDPs on the three components, the differences between the $S_{1-0-4-c}$ and $S_{2-3-4-c}$ scenarios are determined.

**Table 2**. The scenarios to identify the impacts of different clusters of IWDPs on the SHE nexus.

| Different clusters of IWDPs | | The priority orders of S, H, and E | | | Scenarios |
| --- | --- | --- | --- | --- | --- |
| | | S | H | E | |
| Without IWDPs(1) | \ <br> (1-0) | | | | $S_{1-0-4-1}$ |
| | | | ISQ | | $S_{1-0-4-2}$ |
| | | | | | $S_{1-0-4-3}$ |
| | | S-Priority | \ | ISQ | $S_{1-0-1-1}$ |
| | | S-Priority | ISQ | \ | $S_{1-0-1-2}$ |
| | | \ | H-Priority | ISQ | $S_{1-0-2-1}$ |
| | | ISQ | H-Priority | \ | $S_{1-0-2-2}$ |
| | | \ | ISQ | E-Priority | $S_{1-0-3-1}$ |
| | | ISQ | \ | E-Priority | $S_{1-0-3-2}$ |
| With IWDPs(2) | With water donation impacts (2-1) | S-Priority | \ | ISQ | $S_{2-1-1-1}$ |
| | | S-Priority | ISQ | \ | $S_{2-1-1-2}$ |
| | | \ | H-Priority | ISQ | $S_{2-1-2-1}$ |
| | | ISQ | H-Priority | \ | $S_{2-1-2-2}$ |
| | | \ | ISQ | E-Priority | $S_{2-1-3-1}$ |
| | | ISQ | \ | E-Priority | $S_{2-1-3-2}$ |
| | With water receiving impacts (2-2) | S-Priority | \ | ISQ | $S_{2-2-1-1}$ |
| | | S-Priority | ISQ | \ | $S_{2-2-1-2}$ |
| | | \ | H-Priority | ISQ | $S_{2-2-2-1}$ |
| | | ISQ | H-Priority | \ | $S_{2-2-2-2}$ |
| | | \ | ISQ | E-Priority | $S_{2-2-3-1}$ |
| | | ISQ | \ | E-Priority | $S_{2-2-3-2}$ |
| | With water donation and receiving impacts (2-3) | | | | $S_{2-3-4-1}$ |
| | | | ISQ | | $S_{2-3-4-2}$ |
| | | | | | $S_{2-3-4-3}$ |
| | | S-Priority | \ | ISQ | $S_{2-3-1-1}$ |
| | | S-Priority | ISQ | \ | $S_{2-3-1-2}$ |
| | | \ | H-Priority | ISQ | $S_{2-3-2-1}$ |
| | | ISQ | H-Priority | \ | $S_{2-3-2-2}$ |
| | | \ | ISQ | E-Priority | $S_{2-3-3-1}$ |
| | | ISQ | \ | E-Priority | $S_{2-3-3-2}$ |

\* ISQ represents the component is in standard scheduling rules ( i.e., in Status Quo)





## 3 Study area and data

### 3.1 Overview of the study area

The Hanjiang River, as the largest tributary of the Changjiang River, plays an important role in China's economic development and ecological environment (Xia et al., 2020). The HR originates from the Qinling Mountains, and it traverses Shaanxi, Hubei, and Henan before joining the Changjiang River in Wuhan. The Hanjiang River Basin (HRB) has a basin area of about 159,000 km², and has different clusters of IWDPs (Stone and Jia, 2006). In this study, we choose the Han-to-Wei Water Diversion Project (Wei et al., 2020), the Middle Route of the South-to-North Water Diversion Project (Li et al., 2016), and the Northern Hubei Water Resources Allocation Project (He and X, 2020) to analyze the water donation impacts of IWDPs on the SHE nexus. And the Three Gorges Reservoir to Hanjiang River (Yang et al., 2012) and the Changjiang-to-Han River Water Diversion Project (Zhang et al., 2022) are selected to discuss the water receiving impacts in HRB. All IWDPs follow its scheduling rules for donation and receiving. The HRB hosts numerous reservoirs, with 15 cascade reservoirs along its main stream, starting with the Huangjinxia Reservoir. These reservoirs play significant roles in flood control, water supply, hydropower generation, and ecological conservation (Liu et al., 2018). The Huangjinxia Reservoir (HJX), Ankang Reservoir (AK), Danjiangkou Reservoir (DJK), Wangfuzhou Reservoir (WFZ), and Xinglong Reservoir (XL) are chosen as research nodes due to their extensive spatial distribution and different priority orders of S, H, and E. Among them, HJX, DJK, and XL are water supply-prioritized reservoirs, while AK and WFZ are hydropower generation-prioritized reservoirs. The overview map of HRB and the sketch graphic are shown in Figure 4 and 5. The characteristic parameter values of reservoirs are listed in Table 3.

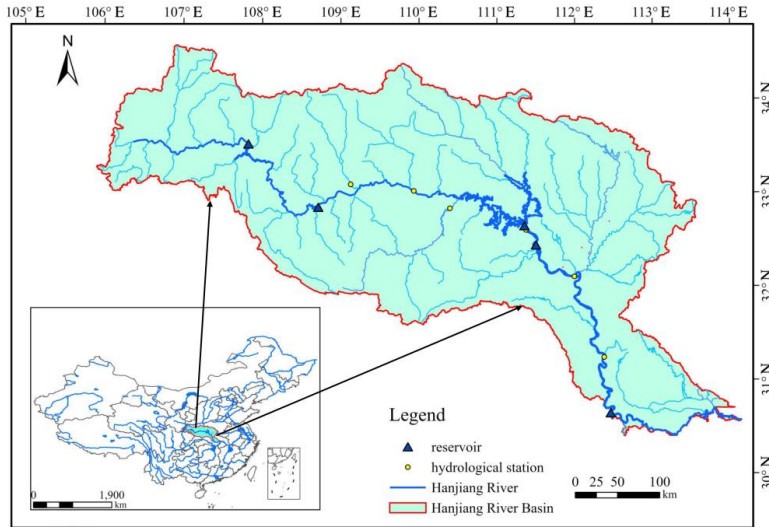

**Figure 4. Overview map of the study area.**





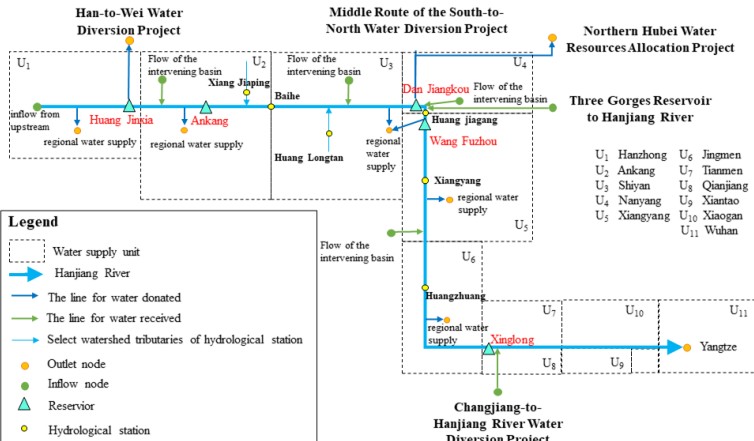

**Figure 5. The sketch graphic of the Hanjiang River Basin (adapted from Zeng et al., 2023).**

**Table 3**. List of characteristic parameter values of reservoirs.

| Characteristic parameter | Unit | HJX | AK | DJK | WFZ | XL |
|---|---|---|---|---|---|---|
| Normal water level | m | 450 | 330 | 170 | 86.23 | 36.2 |
| Usable storage | $10^8 m^3$ | 0.92 | 14.95 | 163.6 | 1.495 | 0.246 |
| Dead water level | m | 440 | 305 | 150 | 85.48 | 35.7 |
| Installed capacity | MW | 135 | 800 | 900 | 109 | 40 |
| Annual generation | billion kW·h | 0.25 | 2.80 | 3.83 | 0.58 | 0.23 |
| Hydropower generation efficiency | / | 8.4 | 8.4 | 7.7 | 8.5 | 8.4 |
| Regulation ability | / | Daily | Yearly | Multi-year | Daily | Daily |

### 3.2 Data sources

Based on the availability of observed runoff data and water supply volume data in the HRB, 1972-2020 is chosen for runoff simulation, and the scenario simulation period is selected as 2006-2020. Observed runoff data was obtained from the Hydrology Bureau of the Changjiang Water Resources Commission, selecting monthly runoff data from six hydrological stations: Xiangjiaping, Baihe, Huanglongtan, Huangjiagang, Xiangyang, and Huangzhuang. Meteorological forcing data for the HRB was sourced from the National Meteorological Science Data Center (http://data.cma.cn/). 88 meteorological stations were selected for the daily precipitation, maximum and minimum temperatures, and average wind speed data from 1972 to 2020. These data were interpolated onto a 5-arc-minute orthogonal grid using the Inverse Distance Weighting method. Digital Elevation Model (DEM) data, with a spatial resolution of 90 meters, was provided by the Geospatial Data Cloud website (http://www.gscloud.cn/). Vegetation parameters data was sourced from the global vegetation cover classification data with 1 km resolution developed by the University of Maryland (http://www.landcover.orgdatalandcover/data.shtml). Soil parameters data was sourced from the Cold and Arid Regions Science Data Center (http://www.bdc.ac.cn/portal/) and utilizes the Harmonized World Soil Database (HWSD) created by the FAO and IIASA, at 5 arc-minute resolution. The relevant physical parameters of soils divided into 14 types including bare soils, were estimated using the Soil-Water Characteristics (SWCT) module in the SPAW software. Reservoir characteristic parameters were primarily sourced from the official websites, reservoir design reports, and related literatures. The water supply volume data was obtained from the "Water Resources Bulletin" of cities in HRB from 2006 to 2020. Based on the water supply data from administrative regions, the water supply volume for the study area is calculated through ArcGIS.





## 4 Results and Discussion

### 4.1 Calibration and verification of VIC model

The HRB was discretized into 2103 grids of 5-arc minutes, and the soil was classfied into three layers for the VIC model. Inputting meteorological forcing, soil parameter, and vegetation parameter data for each grid, runoffs were simulated. Model
warm-up was spanned 1972-1975, while its calibration was conducted from 1976 to 2005, and the validation was from 2006 to 2013. And runoff from 2014 to 2020 was extension simulated for its post-validation. All the results are shown in Figure 6. It can be found that the accuracies of the simulations at all hydrological stations are acceptable, and the superior performances were found in upstream. For instance, $NSE$ for calibration and validation were 0.896 and 0.774, with corresponding $R^2$ of 0.908 and 0.866 at BH. Due to the intense human activity impacts in mid–lower reaches of the HRB, the poorer performance were
found at HJG while their $NSE$ values still exceed 0.600. $PBIAS$ for all these six stations during calibration and validation periods ranged within [-5 %,11 %], which also indicates satisfactory agreement.

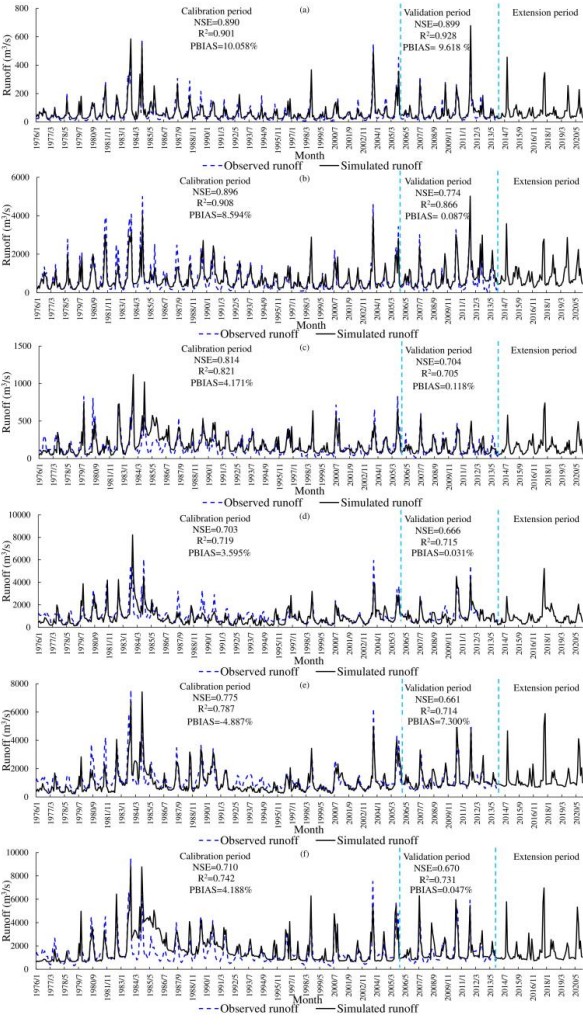

**Figure 6. Calibration and validation results of simulation at hydrological stations: (a)Xiangjiangping Hydrological Station, (b) Baihe Hydrological Station, (c) Huanglongtan Hydrological Station, (d) Huangjiagang Hydrological Station, (e) Xiangyang**





**Hydrological Station, (f) Huangzhuang Hydrological Station.**

**4.2 Multi-level ecological flows classification and calculation results**

The multi-level ecological flows at HJX, AK, DJK, WFZ, and XL reservoir dam sites for each month were determined through the MTMMHC method. Their ecological flows are categorized into four levels: $MEF$, $E_2$, $OEF_{min}$ and $OEF_{max}$. The results at XL reservoir dam site from the MTMMHC method are presented in Table 4. Their ecological flows for wet, normal, and dry

295    years show the decreasing trends, with higher values during the flood season. Its peak ecological flow occurs in August during wet years while in July during both normal and dry years. All the peak ecological flows for the other four sites occur between July and September. The peak ecological flows for HJX and AK reservoir dam sites during wet, normal, and dry years occur between July and August. The peak values for DJK and WFZ are dispersed, and theyare found in September, August, and July. The ecological flows at the five reservoir dam sites from June to September are significantly higher than their in other months.

**Table 4**. Multi-level ecological flows resulted from MTMMHC method.

| Site | Month | Hydrological years | | | | | | | | | | | |
| --- | --- | --- | --- | --- | --- | --- | --- | --- | --- | --- | --- | --- | --- |
| | | Wet year | | | | Normal year | | | | Dry year | | | |
| | | $MEF$ | $E_2$ | $OEF_{min}$ | $OEF_{max}$ | $MEF$ | $E_2$ | $OEF_{min}$ | $OEF_{max}$ | $MEF$ | $E_2$ | $OEF_{min}$ | $OEF_{max}$ |
| XL Reservoir dam site | Jan | 1197 | 1476 | 1550 | 1668 | 825 | 849 | 872 | 910 | 664 | 666 | 668 | 670 |
| | Feb | 1265 | 1467 | 1539 | 1656 | 836 | 863 | 890 | 933 | 675 | 678 | 681 | 686 |
| | Mar | 1268 | 1486 | 1569 | 1702 | 842 | 869 | 896 | 938 | 685 | 690 | 696 | 705 |
| | Apr | 1249 | 1329 | 1426 | 1581 | 868 | 892 | 916 | 955 | 691 | 698 | 704 | 714 |
| | May | 1273 | 1675 | 1822 | 2058 | 861 | 887 | 912 | 953 | 705 | 714 | 723 | 738 |
| | Jun | 1653 | 1681 | 1877 | 2192 | 877 | 916 | 955 | 1017 | 763 | 786 | 809 | 846 |
| | Jul | 1818 | 2629 | 2987 | 3560 | 1288 | 1430 | 1572 | 1799 | 875 | 921 | 968 | 1043 |
| | Aug | 1885 | 2522 | 2849 | 3372 | 1266 | 1401 | 1537 | 1753 | 811 | 845 | 879 | 933 |
| | Sep | 1465 | 2822 | 3225 | 3869 | 1174 | 1279 | 1384 | 1553 | 834 | 879 | 924 | 997 |
| | Oct | 1368 | 2276 | 2611 | 3148 | 978 | 1036 | 1094 | 1186 | 733 | 752 | 772 | 802 |
| | Nov | 1315 | 1586 | 1748 | 2007 | 897 | 932 | 966 | 1022 | 691 | 697 | 704 | 714 |
| | Dec | 1194 | 1471 | 1549 | 1675 | 845 | 873 | 900 | 944 | 680 | 686 | 691 | 700 |

**4.3 Responses of indexes in feedback loops with different clusters of IWDPs in a reservoirs group**

**4.3.1 Responses of indexes in feedback loops without and with IWDPs**

To analyse the feedback loops of SHE nexus without (i.e., $S_{1-0-p-c}$ and $S_{1-0-4-c}$) and with IWDPs (i.e., $S_{2-3-p-c}$ and $S_{2-3-4-c}$) across

the multiple temporal (i.e., monthly, seasonal and annual) and spatial (i.e., five reservoirs) scales, the differences of indexes (i.e., $LRR_1$, $LRR_2$, $LRR_3$ for log response ratio of the S, H, and E component) between $S_{1-0-p-c}$ and $S_{1-0-4-c}$ or between $S_{2-3-p-c}$ and $S_{2-3-4-c}$ are determined at the time scales in a reservoirs group. The results of the monthly differences are shown in Figure 7 and 8.

If there was no IWDPs and S-Priority was set, both the mean values of $LRR_2$ (i.e., -0.062, -0.092, -0.068, -0.094, and -

0.021) and the mean values of $LRR_3$ (i.e., -0.270, -0.539, -0.070, -0.195, and -0.606) in five reservoirs remain below 0 as shown in Figure 7 (a). As there are a large number of negative values of $LRR_2$ in all reservoirs with S-Priority as shown in Figure 7 (a-1), the hydropower generation is found to be reduced in most months. However, there are still some positive values of $LRR_2$ in reservoirs. XL reservoir shows a higher occurrence of positive values of $LRR_2$ when there is abundant water such as July in





2007 and September in 2017 (i.e., 0.145 and 0.123, respectively). As shown in Figure 7 (a-2), all the five reservoirs exhibit a negative $LRR_3$ in all months. The value of $LRR_3$ for the DJK reservoir is closest to 0. The smallest mean values of $LRR_3$ for the XL and AK reservoirs are -0.606 and -0.539, respectively. The reduction of $EFCR_{xy}$ for DJK is smaller than those for other reservoirs due to its effective regulating. The values of $EFCR_{xy}$ for XL and AK significantly decrease due to their greater reductions of ecological flow and their higher ecological flow standards at the two reservoirs' dam sites. The extreme values (e.g., lower than 90 % months values) of $LRR_3$ for HJX, AK, WFZ, and XL reservoirs occur in the higher water supply demand months such as June to September of each year. There are also differences between the results of $LRR_2$ and $LRR_3$, the range of $LRR_3$ value is wider, while its of $LRR_2$ are relatively concentrated and closer to 0. Therefore, there are negative feedbacks of the S component on other two components, and these negative feedbacks of the S component on E are even more pronounced than those on H. Our findings are consistent with the results from the other SHE nexus studies (Khalkhali et al., 2018). It can be also found that the negative feedbacks of S on H in reservoirs are weakened or even broken, while positive feedbacks of S on H are in abundant water months.

If there was no IWDPs and H-Priority was set, the values of $LRR_1$ for all five reservoirs are less than zero in most months, and the mean values of $LRR_3$ exceed zero as shown in Figure 7 (b). The water supply for HJX, DJK, and XL is significantly decreased, with their mean values of $LRR_1$ are -18.345, -11.547, and -7.719, while the water supply for AK and WFZ has slight reductions (i.e., the mean values of $LRR_1$ are -0.162 and -0.225, respectively) as shown in Figure 7 (b-1). There are two positive values of $LRR_1$ for DJK reservoir occurring in January 2010 and in July 2011 (i.e., 20.324 and 0.189, respectively). In January 2010, higher water storage resulting from H-Priority increases water availability. With H-Priority, reservoirs with regulating capacity will store more water, leading to increased generation flow during dry periods (Zhang et al., 2014). While in July 2011, an increase in the discharge flow from the upstream reservoir increase the water supply. As shown in Figure 7 (b-2), the values of $EFCR_{xy}$ for HJX reservoir experiences a significant increase, with a mean value of $LRR_3$ of 0.922, followed by XL and AK (i.e., their mean values of $LRR_3$ are 0.396 and 0.143). DJK and its downstream reservoirs have negative values of $LRR_3$ in abundant water months because of the increased storage capacity and the reduced inflow into DJK. The water resource allocation of DJK affects the SHE system of downstream reservoirs. There are also differences between the results of $LRR_1$ and $LRR_3$, the values of $LRR_3$ are relatively closer to 0 than those of $LRR_1$. The feedbacks on S are more pronounced than on E. The extreme values of $LRR_1$ and $LRR_3$ are always found in months with small water flow in river but with high-water supply demand. Thus, H has both negative and positive feedbacks on E which is consistent with the founding by Wu et al. (2021). In abundant water months, the positive feedback can be changed into a negative one. The increased flow for hydropower generation alleviates the pressure of ecological damage in river. However, the increased discharge inevitably reduces the amount of available water resources for supply, and leads to negative impacts on the S component.

If there was no IWDP and E-Priority was set, the mean values of $LRR_1$ for HJX, DJK, and XL reservoirs are -6.591, -1.740, and -5.643 as shown in Figure 7 (c-1). However, the values of $LRR_1$ for AK and WFZ are almost zero because their increased discharge water from upstream are prioritized to be released for hydropower generation, and no excess is for water supply. Thus, the prioritizing E has less impact on S for reservoirs due to the main function of hydropower generation. DJK and XL exhibit some positive values of $LRR_1$ because the increased inflows from upstream. Therefore, the increased inflow to upstream reservoirs alleviates the negative feedbacks of E on S in downstream reservoirs. As shown in Figure 7 (c-2), the mean values of $LRR_2$ for HJX, AK, DJK, and WFZ reservoirs are 0.127, 0.045, 0.022, and 0.037. While XL has a negative mean value of $LRR_2$ at -0.058, it experiences more decreases in hydropower generation primarily due to its smaller installed capacity (Zhang, 2008). Negative values of $LRR_2$ can be found in abundant water months. The ranges of $LRR_1$ and $LRR_2$ are also different. The former one is wide while the other one is narrow and their values are closer to zero. Therefore, the feedbacks of the E component on S are stronger than those on H. According to the values of $LRR_n$, Negative feedbacks of the E component on S for reservoirs





has been found in the scenario that main function is water supply while no significant effect on reservoirs has been found in the scenario that main function is hydropower generation. There are both negative and positive feedbacks of the E component on H while the negative feedbacks are grown in abundant water months.

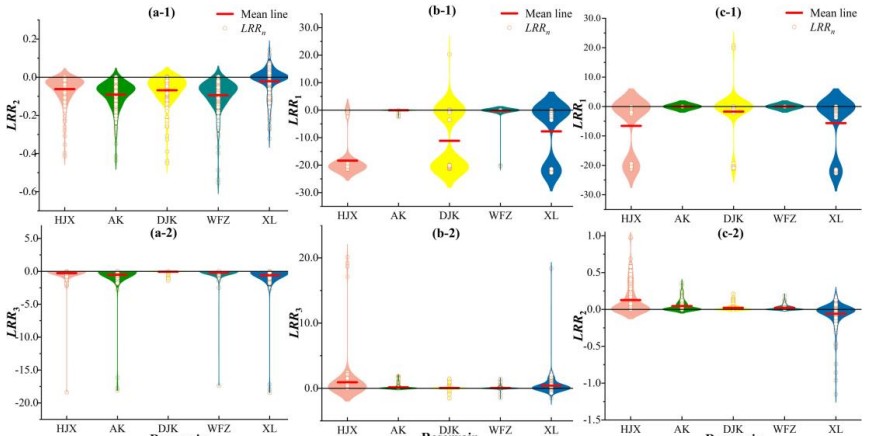

**Figure 7.** $LRR_n$ between $S_{1-0-p-c}$ and $S_{1-0-4-c}$ at the monthly scale: (a-1) $LRR_2$ between $S_{1-0-1-1}$ and $S_{1-0-4-2}$, (a-2) $LRR_3$ between $S_{1-0-1-2}$ and $S_{1-0-4-3}$, (b-1) $LRR_1$ between $S_{1-0-2-1}$ and $S_{1-0-4-1}$, (b-2) $LRR_3$ between $S_{1-0-2-2}$ and $S_{1-0-4-3}$, (c-1) $LRR_1$ between $S_{1-0-3-1}$ and $S_{1-0-4-1}$, (c-2) $LRR_2$ between $S_{1-0-3-2}$ and $S_{1-0-4-2}$.

The differences between the $S_{2-3-p-c}$ and $S_{2-3-4-c}$ scenarios were determined to analyse the feedback loops with IWDPs as shown in Figure 8 (a), (b), and (c). It can be found that the positive or negative signs of the $LRR_n$ values with IWDPs are consistent with those without IWDPs. If there are IWDPs and S-Priority was set, the mean value of $LRR_3$ for XL shows an

increase while all the values of $LRR_2$ and $LRR_3$ for other four reservoirs are lower than those without IWDPs as shown in Figure 8 (a) and Figure 7 (a). The mean values of $LRR_2$ with IWDPs for the five reservoirs are -0.130, -0.114, -0.165, -0.209, and -0.066, and the mean values of $LRR_3$ are -0.908, -0.753, -1.253, -1.125, and -0.285. And DJK reservoir get more extreme values due to the impacts of IWDPs. The values of $LRR_2$ with IWDPs are lower than -0.450 (i.e., the minimum value of $LRR_2$ without IWDPs) in 6 % of the months while the values of $LRR_3$ are lower than -1.404 (i.e., the minimum value of $LRR_3$ without IWDPs)

in 8 % of the months. It is evident that IWDPs strengthens the negative feedbacks of the S component on the other two components in HJX, AK, DJK and WFZ, while IWDPs weaken negative feedbacks of S on E for XL. As shown in Figure 8 (b-1), If there were IWDPs and H-Priority was set, the mean values of $LRR_1$ for HJX, AK, and XL reservoirs significantly decrease to -18.777, -0.783, and -12.242, but the mean value of $LRR_1$ for DJK reservoir are increased by 3.491 due to IWDPs. The operation of the Han-to-Wei Water Diversion Project, the Middle Route of the South-to-North Water Diversion Project, and the

Northern Hubei Water Resources Allocation Project in DJK and upstream reservoirs have reduced the regional water supply (Hong et al., 2016), the differences of water supply between the $S_{2-3-2-c}$ and $S_{2-3-4-c}$ scenarios remain negligible despite further reductions in water supply with H-Priority. As shown in Figure 8 (b-2), The values of $LRR_3$ for HJX, AK, DJK, and WFZ increase further than them in Figure 7 (b-2) without IWDPs, indicating the positive feedbacks of the H component on E get strengthen with the impacts of IWDPs. The values of $LRR_3$ for XL decrease slightly due to the positive feedbacks of the H

component on E and the IWDPs impacts. As shown in Figure 8 (c-1), If there were IWDPs and E-Priority was set, the mean values of $LRR_1$ for HJX and XL decrease by 5.107 and 2.766, respectively. And the mean values of $LRR_1$ for AK and WFZ remain at almost zero, while the mean value of $LRR_1$ for DJK increases by 0.259 with IWDPs compared to without IWDPs. As shown in Figure 8 (c-2), the mean values of $LRR_2$ for five reservoirs increase by 0.176, 0.036, 0.031, 0.021 and 0.008 with IWDPs compared to without IWDPs. The positive feedbacks of E component on H are strengthened, while the negative



feedbacks are weakened.

Therefore, negative feedbacks can be found between S and H, and between S and E while positive feedbacks can be found between H and E in a reservoirs group without IWDPs. These negative and positive feedbacks in our study have also been found in other studies on the SHE nexus (Doummar et al., 2009; Wu et al., 2022). As our proposed framework is valid, the results also reinforce the robustness of the identified feedbacks in different contexts. It has been found that there are a few

positive feedbacks between S and H in abundant water months even the abandoned water leads to a reduction in hydropower generation (Jiang et al., 2018). Thus, the increasing water storage or increasing water supply still can ensure hydropower generation. However, the positive feedbacks between H and E are weakened or even turn to be negative in the small installed hydropower generation capacity reservoirs (e.g., the XL reservoir) even in abundant water months, particularly. The negative feedbacks between S and H, and between S and E are strong in low flow months due to the high-water supply demand. More

competitions for water can be found among S, H and E in low flow months, and their negative feedbacks of the SHE nexus have found to be strengthened. Feedback loops of SHE nexus in reservoirs with regulation function (e.g., AK and DJK) remain stable under the varying inflow conditions. These reservoirs reasonably allocate water among S, H and E components to prevent strengthening of negative feedbacks in low flow months. Furthermore, increasing hydropower generation flow might have impacts on downstream water quality and biodiversity (Botelho et al., 2017; Martinez et al., 2019), the feedbacks of H on E are

enhanced. If there were IWDPs, it is evident that feedback loops of SHE nexus across different spatial scales exhibit strong responses. As IWDPs export or import water to or from an area, the amount of available water has to be altered. It can prompt a redistribution and re-planning of the available water (Li, et al., 2014). And the redistribution and re-planning can significantly impact on feedback loops of SHE nexus. Although strong responses occur in feedback loops of SHE nexus, its positive or negative nature of feedback among these components remains stable with impacts of IWDPs. Thus, the redistribution and re-

planning of available water can not alter their competitions and collaborations among the components of the SHE nexus.

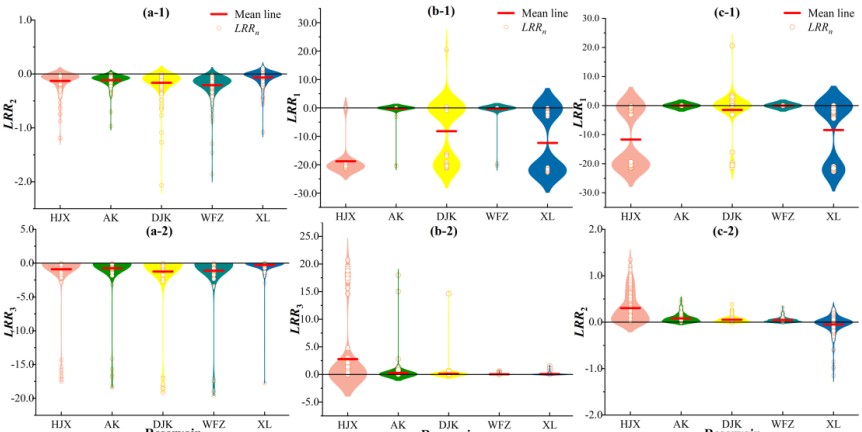

**Figure 8.** $LRR_n$ between $S_{2-3-p-c}$ and $S_{2-3-4-c}$ at the monthly scale: (a-1) $LRR_2$ between $S_{2-3-1-1}$ and $S_{2-3-4-2}$, (a-2) $LRR_3$ between $S_{2-3-1-2}$ and $S_{2-3-4-3}$, (b-1) $LRR_1$ between $S_{2-3-2-1}$ and $S_{2-3-4-1}$, (b-2) $LRR_3$ between $S_{2-3-2-2}$ and $S_{2-3-4-3}$, (c-1) $LRR_1$ between $S_{2-3-3-1}$ and $S_{2-3-4-1}$, (c-2) $LRR_2$ between $S_{2-3-3-2}$ and $S_{2-3-4-2}$.

In this study, March, April, May are taken as spring, June, July and August are taken as summer, September, October and November are taken as autumn, and December, January and February of the following year are taken as winter. The values of $LRR_n$ for five reservoirs at seasonal scale are shown in Figure 9. If there was no IWDP but S-Priority was still set, positive values of $LRR_2$ for HJX and XL are found in summer, while all negative values of $LRR_2$ for other three reservoirs are found in all seasons as shown in Figure 9 (a). The mean values of $LRR_3$ for the five reservoirs are -0.119, -0.106, -0.022, -0.020, and -

0.669, and all values of $LRR_3$ are negative in all seasons. If there were IWDPs and S-Priority was set, the mean value of $LRR_3$




for XL increases while the values of $LRR_2$ and $LRR_3$ for other four reservoirs are less than those without IWDPs as shown in Figure 9 (b). These negative values indicate that IWDPs significantly strengthen the negative feedbacks of the S component on H and E in reservoirs and weaken negative feedback of S on E in XL. If there was no IWDPs but H-Priority was set, negative values of $LRR_1$ and positive values of $LRR_3$ are found for the five reservoirs as shown in Figure 9 (c). For HJX, DJK and XL

reservoirs, the negative values of $LRR_1$ are found in winter while zero values of $LRR_1$ are found in summer. The mean values of $LRR_1$ are close to zero in AK and WFZ reservoirs in all seasons. Positive values of $LRR_3$ are smaller in HJX, AK, DJK and WFZ reservoirs, while those in XL are greater in winter with a low flow. If there were IWDPs and H-Priority was set, the values of $LRR_1$ for all reservoirs are lower than those without IWDPs as shown in Figure 9 (d). Values of $LRR_3$ for HJX, AK, DJK and WFZ reservoirs are greater than those without IWDPs, while those for XL are close to zero. If there was no IWDPs and E-

Priority was set, negative values of $LRR_1$ for HJX, DJK, WFZ and XL reservoirs can be found in almost every season, while zero values of $LRR_1$ for AK reservoir can be found in all seasons. As shown in Figure 9 (e), two positive values of $LRR_1$ for DJK are found in spring and in winter of 2007 due to the increased discharge water from AK reservoir. The positive values of $LRR_2$ for the five reservoirs are found in most seasons, but few negative values are found in summer. If there were IWDPs and E-Priority was set, more positive values of $LRR_2$ for five reservoirs and less negative values of $LRR_1$ are found in HJX, DJK,

WFZ and XL reservoirs. Therefore, negative feedbacks can be found between S and H, and between S and E while positive feedbacks can be found between H and E in most seasons in a reservoirs group. These feedbacks are strengthened in winter, while positive feedbacks between S and H and negative feedbacks between H and E are found in summer. IWDPs strongly impact these feedback loops, but the positive or negative nature of feedbacks among SHE remains stable at seasonal scale.

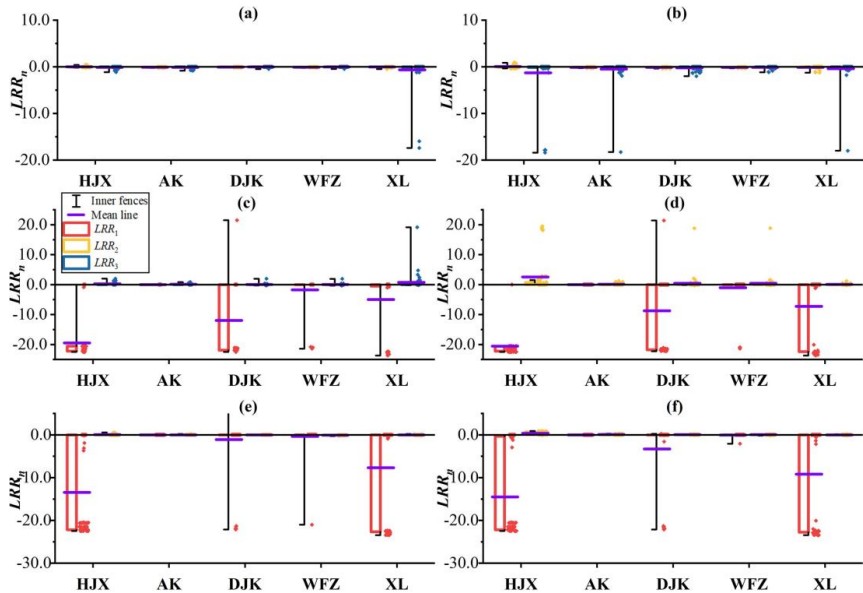

**Figure 9.** $LRR_n$ **between S$_{1-0-p-c}$ and S$_{1-0-4-c}$ and between S$_{2-3-p-c}$ and S$_{2-3-4-c}$ at seasonal scale: (a) $LRR_n$ between S$_{1-0-1-c}$ and S$_{1-0-4-c}$, (b)** $LRR_n$ **between S$_{2-3-1-c}$ and S$_{2-3-4-c}$, (c) $LRR_n$ between S$_{1-0-2-c}$ and S$_{1-0-4-c}$, (d) $LRR_n$ between S$_{2-3-2-c}$ and S$_{2-3-4-c}$ (e) $LRR_n$ between S$_{1-0-3-c}$ and S$_{1-0-4-c}$, (f) $LRR_n$ between S$_{2-3-3-c}$ and S$_{2-3-4-c}$.**

The values of $LRR_n$ for five reservoirs at annual scale are shown in Figure 10. If there was no IWDPs and S-Priority was set, values of $LRR_2$ for HJX, AK, WFZ reservoirs are negative during 2006-2020 as shown in Figure 10 (a-1). There are two

positive values of $LRR_2$ for DJK in 2010, 2018, and one positive values for XL in 2020. And there is abundant water in all these three years. The minimum values of $LRR_2$ for five reservoirs are both found in the driest year. And there are more small values





in AK and WFZ. The mean values of $LRR_3$ for five reservoirs are -0.020, -0.026, -0.034, -0.058, and -0.062 as shown in Figure 10 (a-2). The small values of $LRR_3$ for five reservoirs are found in dry years or high ecological flow requirement years such as 2010, 2011 and 2017. Downstream reservoirs can bring stronger negative feedbacks of S on E, so WFZ and XL have more

small values of $LRR_3$. If there was no IWDPs but H-Priority was still set, the zero values of $LRR_1$ for AK and WFZ are found in all years, and WFZ gets more negative values of $LRR_1$. The positive values of $LRR_3$ for five reservoirs are found in abundant water years as shown in Figure 10(b-2), while negative values of $LRR_2$ for DJK and its upstream reservoirs are found because of the increased water storage from DJK in these years. If there was no IWDPs but E-Priority was still set, negative values of $LRR_1$ for HJX, DJK and XL and the positive values of $LRR_2$ can be found in dry years and high ecological flow requirement

years as shown in Figure 10 (c-1). The negative values of $LRR_2$ are mainly found in abundant water years as shown in Figure 10 (c-2). As shown in Figure 10 (d), (e), (f), negative and positive values of $LRR_n$ for HJX, AK, DJK, WFZ, and values of $LRR_1$, $LRR_2$ for XL turn to be more extreme than those without IWDPs. The values of $LRR_3$ for XL are closer to zero if there were IWDPs.

  Therefore, signs of mean values of $LRR_n$ at seasonal and annual scales are consistent with those at monthly scale, so the

feedback loops of SHE nexus exhibit intrinsic similarity and stability across different time scales. Compared with the values of $LRR_n$ at monthly scale, the values at the seasonal scale show its stronger periodic variations. These periodic variations align closely with the runoff variations, and the temporal and spatial variations in feedback loops are primarily attributed to variations in runoff. Since the seasonal runoff changes are found to be more than those at a monthly scale (Xu et al., 2018), the seasonal results can help analyze the variations in periodic feedback loops. Compared with monthly and seasonal scales, results at the

annual scale reveal the long-term trends and periodic variations in the inter-annual and spatial trends of the SHE nexus from a macro perspective. Compared with seasonal and annual scales, the impacts of reservoir operation and the regulation on SHE nexus can be clearly simulated and observed at the monthly scale, while the immediate changes and variations in the nexus at monthly scale can provide information for short-term decision-making in reservoirs.



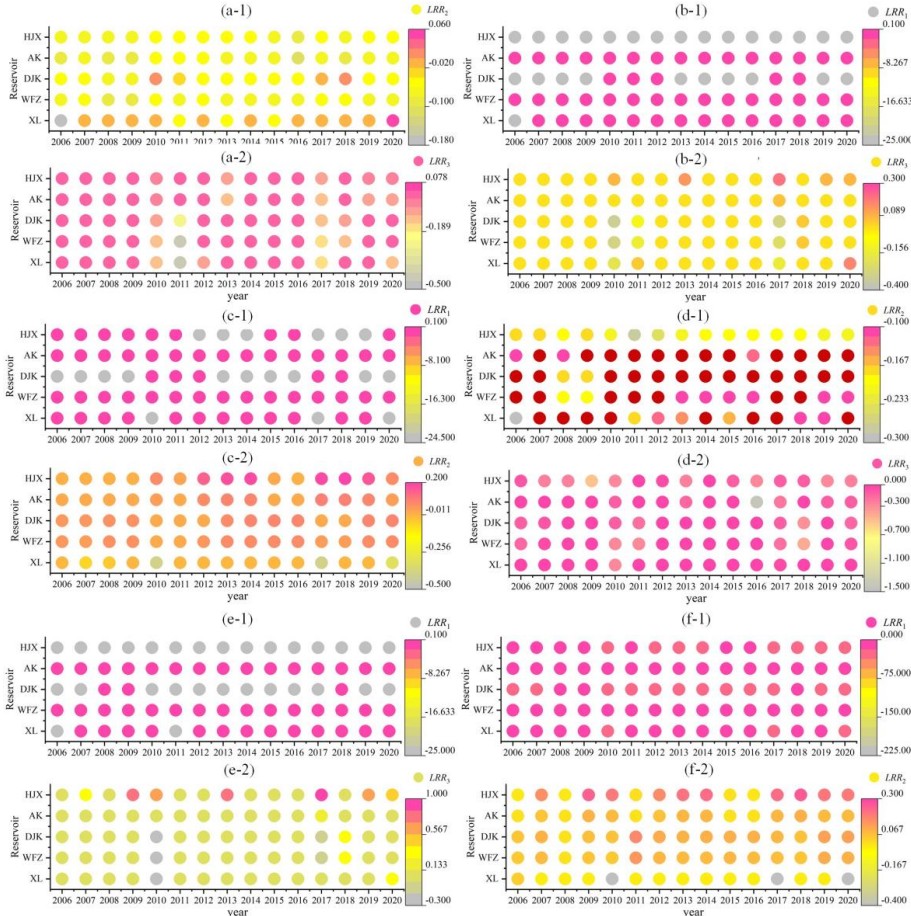

**Figure 10.** $LRR_n$ between $S_{1-0-p-c}$ and $S_{1-0-4-c}$ and between $S_{2-3-p-c}$ and $S_{2-3-4-c}$ at annual scale: (a-1) $LRR_2$ between $S_{1-0-1-1}$ and $S_{1-0-4-2}$, (a-2) $LRR_3$ between $S_{1-0-1-2}$ and $S_{1-0-4-3}$, (b-1) $LRR_1$ between $S_{1-0-2-1}$ and $S_{1-0-4-1}$, (b-2) $LRR_3$ between $S_{1-0-2-2}$ and $S_{1-0-4-3}$, (c-1) $LRR_1$ between $S_{1-0-3-1}$ and $S_{1-0-4-1}$, (c-2) $LRR_2$ between $S_{1-0-3-2}$ and $S_{1-0-4-2}$, (d-1) $LRR_2$ between $S_{2-3-1-1}$ and $S_{2-3-4-2}$, (d-2) $LRR_3$ between $S_{2-3-1-2}$ and $S_{2-3-4-3}$, (e-1) $LRR_1$ between $S_{2-3-2-1}$ and $S_{2-3-4-1}$, (e-2) $LRR_3$ between $S_{2-3-2-2}$ and $S_{2-3-4-3}$, (f-1) $LRR_1$ between $S_{2-3-3-1}$ and $S_{2-3-4-1}$, (f-2) $LRR_2$ between $S_{2-3-3-2}$ and $S_{2-3-4-2}$.

**4.3.2 Responses of indexes in feedback loops with only water donation, water receiving, and both donation and receiving**

To analyse the impacts of only water donation (i.e., $S_{2-1-p-c}$ and $S_{1-0-4-c}$), only water receiving (i.e., $S_{2-2-p-c}$ and $S_{1-0-4-c}$), and both donation and receiving (i.e., $S_{2-3-p-c}$ and $S_{1-0-4-c}$) on feedback loops of SHE nexus across the multiple temporal and spatial scales, the differences of indexes between $S_{2-m-p-c}$ and $S_{1-0-4-c}$ are determined in a reservoirs group. The results of the monthly differences are shown in Figure 11-13.

If there was only water donation and S-Priority was set, values of $LRR_2$ and $LRR_3$ for five reservoirs are negative and lower than those without IWDPs as shown in Figure 11 (a-1) and (a-2). More small negative values are found in DJK, water donation has negative impacts on the negative feedback of S on H and E for five reservoirs. If there was only water receiving and S-Priority was set, values of $LRR_2$ and $LRR_3$ for HJX and AK are the same as those without IWDPs. Meanwhile, for DJK, WFZ, and XL, the values are close to zero. XL exhibits a lot of positive values of $LRR_3$ as shown in Figure 11 (b-1) and (b-2). If there were both water donation and receiving, the mean values of $LRR_2$ for five reservoirs are -0.594, -0.263, -0.484, -0.468 and -0.091, and mean values of $LRR_3$ for five reservoirs are -6.117, -1.500, -2.011, -1.598 and 0.143 as shown in Figure 11 (c-1) and

(c-2). There are negative impacts on negative feedbacks of S on H and E for HJX, AK, DJK and WFZ and positive impacts of the negative feedbacks of S on E for XL.

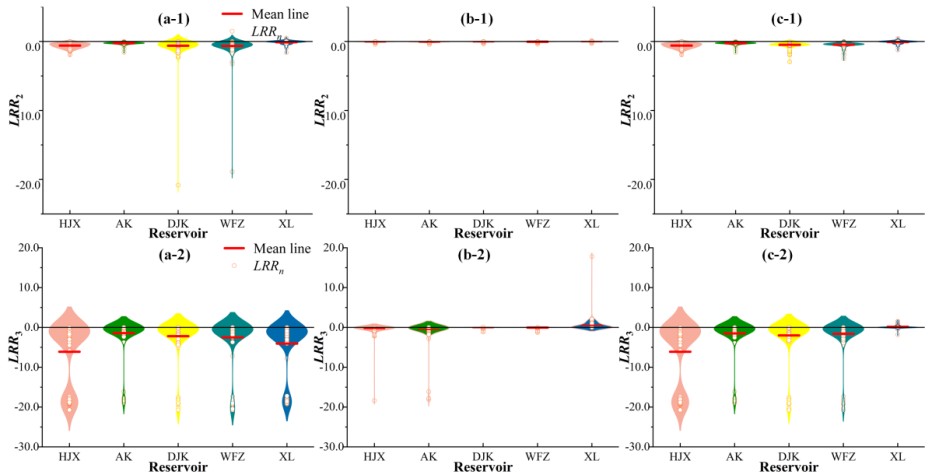

Figure 11. $LRR_n$ values of five reservoirs when there are different clusters of IWDPs and S-Priority was set at the monthly scale: (a-1) and (a-2) are $LRR_2$ and $LRR_3$ when there is only water donation, (b-1) and (b-2) are $LRR_2$ and $LRR_3$ when there is only water receiving, (c-1) and (c-2) are $LRR_2$ and $LRR_3$ when there are both donation and receiving.

If there was only water donation and H-Priority was set, values of $LRR_1$ and $LRR_3$ for five reservoirs are lower than those without IWDPs as shown in Figure 12 (a-1) and (a-2). Negative values of $LRR_3$ for five reservoirs are found in low flow months such as November, December and January. Thus, water donation is found to have negative impacts on feedbacks of H on S and E, especially in low flow months. If there was only water receiving and H-Priority was set, values of $LRR_1$ and $LRR_3$ for DJK, WFZ and XL are greater than those without IWDPs as shown in Figure 12 (b-1) and (b-2). Water receiving has positive impacts on feedbacks of H on S and E. If there were both water donation and receiving and H-Priority was set, the mean values of $LRR_1$ and $LRR_3$ for DJK, WFZ and XL are still lower than those without IWDPs. And the mean value of $LRR_3$ for XL is greater than those without IWDPs as shown in Figure 12 (c-1) and (c-2).

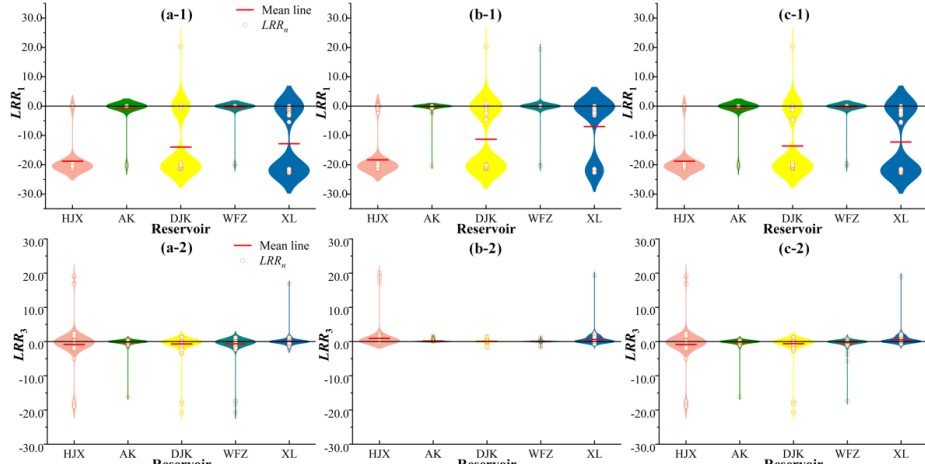

Figure 12. $LRR_n$ values of five reservoirs when there are different clusters of IWDPs and H-Priority was set at the monthly scale: (a-1) and (a-2) are $LRR_2$ and $LRR_3$ when there is only water donation, (b-1) and (b-2) are $LRR_2$ and $LRR_3$ when there is only water receiving, (c-1) and (c-2) are $LRR_2$ and $LRR_3$ when there are both donation and receiving.





If there was only water donation and E-Priority was set, then values of $LRR_1$ and $LRR_2$ for five reservoirs are shown in Figure 13 (a-1) and (a-2). The mean values of $LRR_1$ for these five reservoirs are -11.699, -0.002, -7.228, -0.218, and -9.139, respectively. And the mean values of $LRR_2$ are -0.161, -0.067, -0.287, -0.296, and -0.083. All these values are lower than the those without IWDPs as shown in Figure 7 (c-1) and (c-2). Different from the values of $LRR_n$ without IWDPs, there are no positive values of $LRR_1$ for DJK and few positive values of $LRR_2$ for five reservoirs due to the decreased inflows from upstream

with water donation. If there was only water receiving and E-Priority was set, values of $LRR_1$ and $LRR_2$ for DJK, WFZ and XL are greater than those without IWDPs. If there were both water donation and receiving and E-Priority was set, the mean values of $LRR_1$ and $LRR_2$ for DJK, WFZ and XL are still lower than those without IWDPs as shown in Figure 13 (c-1) and (c-2).

Therefore, it is evident that water donation has negative impacts on the negative feedbacks between S and H, on the negative feedbacks between S and E, and on the positive feedbacks between H and E while receiving water has positive impacts

on all these feedbacks. Water donation results in a reduction of available water (Mok et al., 2015) and leads to lower flow. More competition for water can be found among S, H and E, and negatively impacts on the feedbacks. Less competition is found among S, H and E in water receiving areas, and it has positive impacts on their feedbacks.

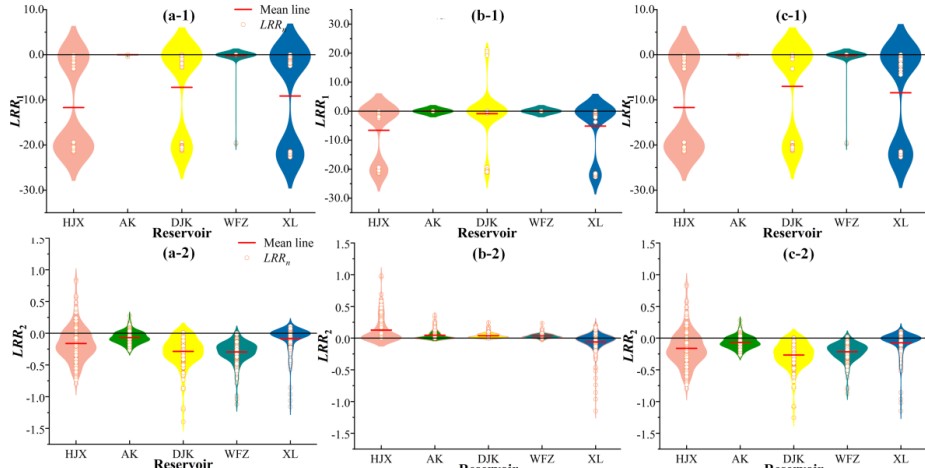

**Figure 13. $LRR_n$ values of five reservoirs when there are different clusters of IWDPs and E-Priority was set at the monthly scale: (a-**
**1) and (a-2) are $LRR_1$ and $LRR_2$ when there is only water donation, (b-1) and (b-2) are $LRR_1$ and $LRR_2$ when there is only water receiving, (c-1) and (c-2) are $LRR_1$ and $LRR_2$ when there are both donation and receiving.**

If there was only water donation and S-Priority was set, values of $LRR_2$ and $LRR_3$ as shown in Figure 14(a-1) are lower than those without IWDPs in all seasons as shown in Figure 9 (a). If there was only water receiving and S-Priority was set, mean values of $LRR_2$ and $LRR_3$ for DJK, WFZ and XL are -0.040, -0.045, -0.026 and -0.012, -0.002, 0.703 as shown in Figure

14 (a-2), and these values are all greater than those without IWDPs. If there were both water donation and receiving and S-Priority was set, mean values of $LRR_2$ for five reservoirs decrease by 0.334, 0.118, 0.336, 0.362 and 0.074 compared to those without IWDPs. Mean values of $LRR_3$ for HJX, AK, DJK and WFZ decrease by 3.692, 0.520, 0.724, 0.550, and its for XL increases by 0.894 compared to those without IWDPs as shown in Figure 14 (a-3). If there was only water donation and H-Priority was set, values of $LRR_1$ and $LRR_3$ as shown in Figure 14(b-1) are lower than those without IWDPs as shown in Figure

9 (c). Mean values of $LRR_1$ for five reservoirs are -20.579, 0, -14.490, -1.752 and -8.124. Mean values of $LRR_3$ for five reservoirs are 0.008, 0.010, -0.073, -0.055 and 0.667. Water donation has negative impacts on feedbacks of H on S for HJX, DJK and XL. If there was only water receiving and H-Priority was set, mean values of $LRR_2$ for DJK, WFZ and XL increase by 0.730, 0.318 and 0.729, and mean values of $LRR_3$ for DJK, WFZ and XL increase by 0, 0.009 and 0.006 compared to those without IWDPs as shown in Figure 14 (b-2). If there were both water donation and receiving and H-Priority was set, mean values of $LRR_2$ for

five reservoirs are -20.579, 0, -14.490, -1.752, -8.068, and mean values of $LRR_3$ for five reservoirs are 0.008, 0.010, -0.050, -0.022 and 0.680 as shown in Figure 14 (b-3). If there was only water donation and E-Priority was set, it can be found that values of $LRR_1$ and $LRR_2$ in all seasons, as shown in Figure 9 (e), are lower than those without IWDPs as shown in Figure 14(c-1). Mean values of $LRR_1$ for five reservoirs decrease by 14.581, 0.010, 9.392, 1.043 and 10.376, and mean values of $LRR_2$ for five reservoirs decrease by 0.054, 0.043, 0.277, 0.331 and 0.221. Water donation has negative impacts on the feedbacks of E on S

and H. If there was only water receiving and E-Priority was set, mean values of $LRR_1$ and $LRR_2$ for DJK, WFZ and mean values of $LRR_1$ for XL are greater than those without IWDPs, while mean values of $LRR_2$ for XL get an increase as shown in Figure 14 (c-2). If there were both water donation and receiving and E-Priority was set, mean values of $LRR_1$ for five reservoirs are -14.518, 0, -9.050, -0.731 and -9.654 and mean values of $LRR_2$ for five reservoirs are -0.002, -0.018, -0.244, -0.200 and -0.002 as shown in Figure 14 (c-3). Values of $LRR_1$ and $LRR_2$ for DJK and WFZ and values of $LRR_1$ foe XL are greater than those with

only water donation, while lower than those without IWDPs. While values of $LRR_2$ for XL are greater than those without IWDPs because of the reduced abandoned water. Therefore, values of $LRR_n$ at seasonal scale demonstrate a consistent conclusion with those at the monthly scale. Moreover, the values of $LRR_n$ are relatively stable in summer, while they change greatly in winter at seasonal scale. The impacts of IWDPs on SHE nexus are more significant in low flow seasons.

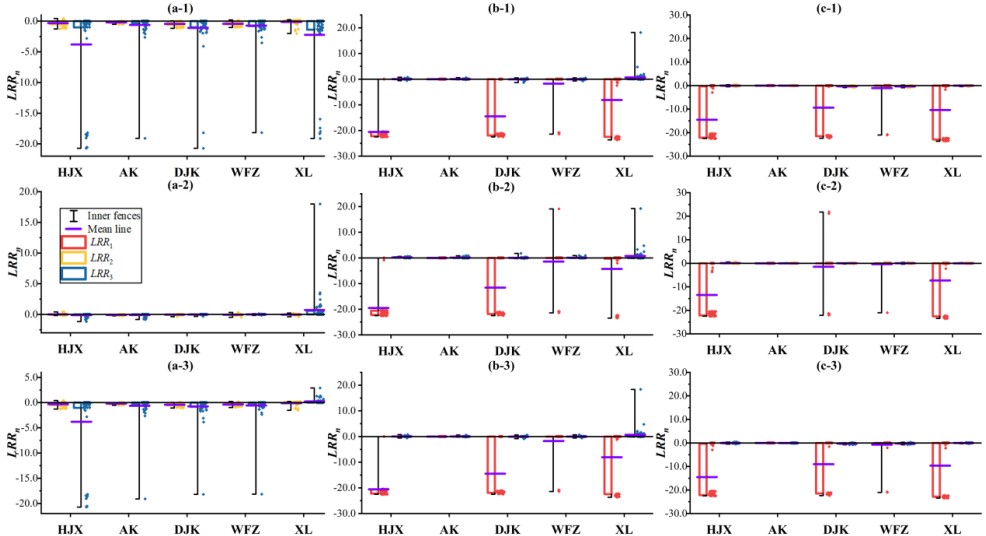

**Figure 14. $LRR_n$ values of five reservoirs when there are different clusters of IWDPs at the seasonal scale: (a-1), (a-2) and (a-3) are $LRR_n$ when there was only water donation, when there was only water receiving, when there were both donation and receiving and S-Priority was set; (b-1), (b-2) and (b-3) are those when H-Priority was set; (c-1), (c-2) and (c-3) are those when E-Priority was set.**

    The results of the annual differences are shown in Figure 15-17. If there was only water donation and S-Priority was set, values of $LRR_2$ and $LRR_3$ are lower than those without IWDPs as shown in Figure 10 (a-1) and (a-2). The values of $LRR_2$ and

$LRR_3$ for HJX, DJK and XL decrease significantly, and these three reservoirs are severely impacted by water donation. If there was only water receiving and S-Priority was set, values of $LRR_2$ and $LRR_3$ for DJK, WFZ and XL show a slight increase. If there were both water donation and receiving and S-Priority was set, only XL has greater values of $LRR_2$ and $LRR_3$ than those without IWDPs.

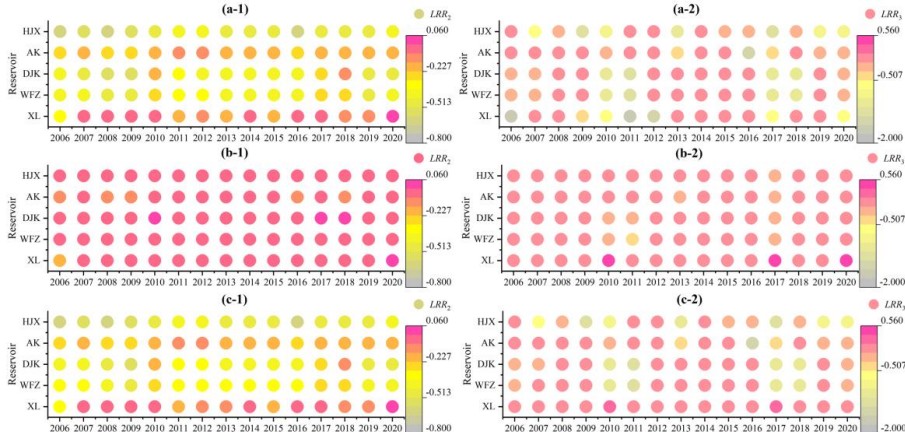

**Figure 15.** $LRR_n$ values of five reservoirs when there are different clusters of IWDPs and S-Priority was set at the annual scale: (a-1) and (a-2) are $LRR_2$ and $LRR_3$ when there was only water donation, (b-1) and (b-2) are those when there was only water receiving, (c-1) and (c-2) are those when there were both donation and receiving.

If there was only water donation and H-Priority was set, HJX, DJK and XL have more negative values of $LRR_1$ as shown in Figure 16 (a-1), and all of these values are lower than those without IWDPs. DJK, WFZ and XL has more smaller values of $LRR_3$ as shown in Figure 16(a-2) than those without IWDPs as shown in Figure 10 (b-2). Smaller values of $LRR_1$ and $LRR_3$ for reservoirs are found in low flow years. If there was only water receiving and H-Priority was set, values of $LRR_1$ and $LRR_3$ for DJK, WFZ and XL increase only in low flow years as shown in Figure 16 (b-1) and (b-2). If there were both water donation and receiving and H-Priority was set, values of $LRR_3$ for XL are greater than those without IWDPs, while all other values of $LRR_1$ and $LRR_3$ are lower than those without IWDPs as shown in Figure 16(c-1) and (c-2).

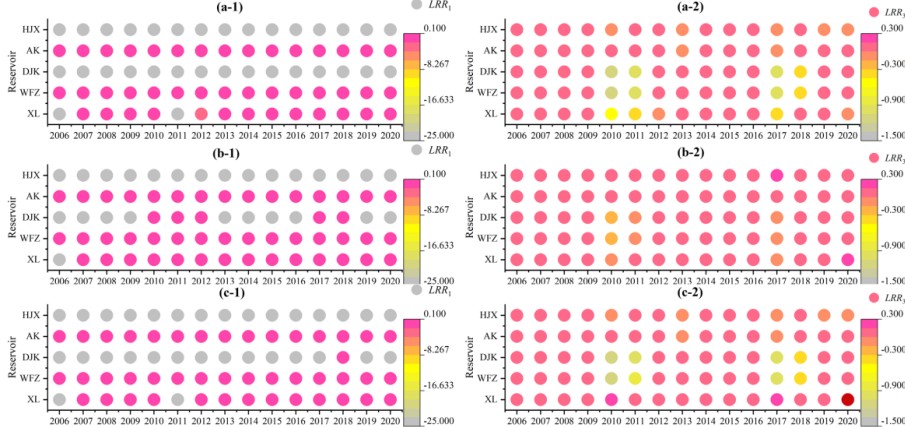

**Figure 16.** $LRR_n$ values of five reservoirs when there are different clusters of IWDPs and H-Priority was set at the annual scale: (a-1) and (a-2) are $LRR_2$ and $LRR_3$ when there was only water donation, (b-1) and (b-2) are those when there was only water receiving, (c-1) and (c-2) are those when there were both donation and receiving.

If there was only water donation and E-Priority was set, more negative values of $LRR_1$ for HJX, DJK and XL are found in low flow years as shown in Figure 17 (a-1) and all of these values are lower than those without IWDPs as shown in Figure 10 (c-1). All five reservoirs get more smaller values of $LRR_2$ and only value of $LRR_2$ for XL in 2007 and 2008 increase as shown in Figure 17 (a-2) because of the reduced abandoned water with water donation. If there was only water receiving and E-Priority was set, there are no change on values of $LRR_1$ for five reservoirs as shown in Figure 17 (b-1), so water receiving has minimal




impact on feedbacks of E on S. values of $LRR_2$ for DJK, WFZ and XL are greater than those without IWDPs. If there were both

water donation and receiving and H-Priority was set, values of $LRR_1$ for HJX, DJK and XL are found to be similar to those with only water donation. Values of $LRR_2$ for DJK and WFZ are greater than those with only water receiving.

Therefore, water donation has negative impacts on the negative feedbacks between S and H, on the negative feedbacks between S and E, and on the positive feedbacks between H and E, while receiving water has positive impacts on these feedbacks across different time scales. Compared with the values of $LRR_n$ at monthly scale, the values of $LRR_n$ at seasonal and annual

scales are stable and changes can be found in low flow periods.

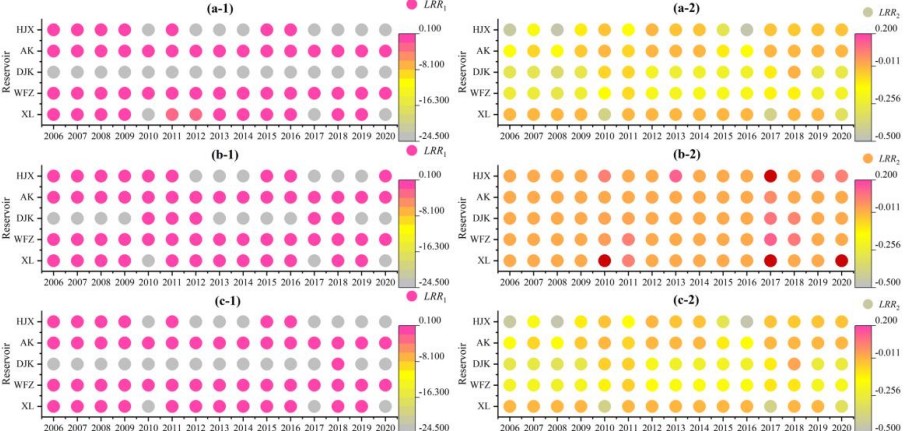

**Figure 17.** *$LRR_n$ values of five reservoirs when there are different clusters of IWDPs and E-Priority was set at the annual scale: (a-1) and (a-2) are $LRR_2$ and $LRR_3$ when there was only water donation, (b-1) and (b-2) are those when there was only water receiving, (c-1) and (c-2) are those when there were both donation and receiving.*

**4.4 Responses of the three components with IWDPs**

To identify the impacts of IWDPs on S, H and E components in a reservoirs group, differences between indexes without IWDPs and with IWDPs (i.e., $S_{2-3-4-c}$ and $S_{1-0-4-c}$) are determined. Negative values of $LRR_1$ for five reservoirs are found in all months, mean values of $LRR_1$ for five reservoirs are -0.002, -0.002, -5.540, -0.218 and -0.013 as shown in Figure 18 (a). It is found that values of $LRR_1$ for DJK are significantly smaller than those for other reservoirs. These IWDPs have notable negative impacts

on the water supply from DJK. Smaller values of $LRR_1$ for DJK and WFZ primarily are found in low flow months. There are negative values of $LRR_1$ for five reservoirs are found in most months, while some positive values are found in abundant water months. Mean values of $LRR_2$ for five reservoirs are -0.464, -0.149, -0.320, -0.259 and -0.025 as shown in Figure 18 (b). So IWDPs have negative impacts on hydropower generation, but they have positive impacts on H in abundant water months. Positive values of $LRR_3$ are found in XL and negative values of $LRR_3$ are found in HJX, AK, DJK and WFZ in all months, mean

values of $LRR_3$ for five reservoirs are -5.208, -0.747, -0.758, -0.473 and 0.428 as shown in Figure 18 (c). There are water donations for the Han-to-Wei Water Diversion Project, the Middle Route of the South-to-North Water Diversion Project and the Northern Hubei Water Resources Allocation Project. All these three IWDPs have negative impacts on the water supply, hydropower generation and environment conservation in HJB. Water receiving from the Three Gorges Reservoir to Hanjiang River are not compensate for all their negative impacts. Water receiving from the Changjiang-to-Hanjiang River Water

Diversion Project benefits environment conservation for XL.

Therefore, S, H and E for all reservoirs are impacted by IWDPs. Water donation results in a reduction of available water for water donation areas, so it has negative impacts on water supply, hydropower generation and environment conservation form these areas, while water receiving has positive impacts on S, H and E for water receiving areas because of increased





available water.

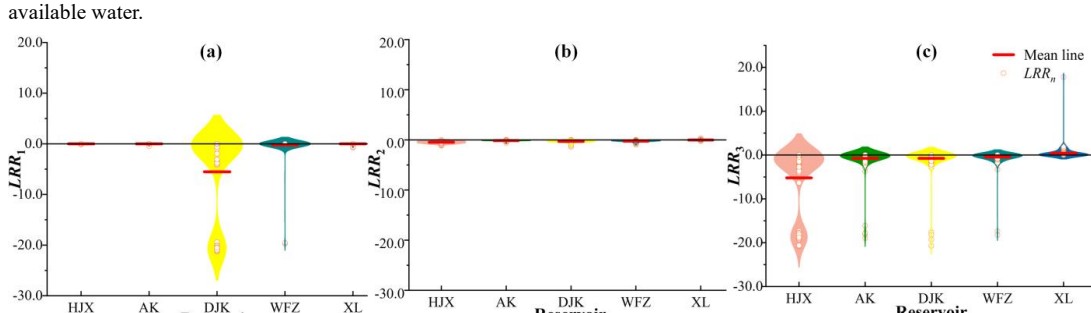

**Figure 18. (a)**$LRR_1$**, (b)**$LRR_2$** and (c)**$LRR_3$** between S**$_{2\text{-}3\text{-}4\text{-}c}$** and S**$_{1\text{-}0\text{-}4\text{-}c}$** at the monthly scale.**

## 5 Conclusions

A framework was proposed to address the different impacts of IWDPs on the dynamic SHE nexus across the multiple temporal and spatial scales in reservoirs group with different priority functions, and to explore collaborative states in feedback loops. The HRB was taken as case study to verify the feasibility and reliability of this framework. Negative feedbacks can be found between S and H, and between S and E while positive feedbacks can be found between H and E in a reservoirs group without IWDPs. The negative feedbacks of S on H and the positive feedbacks of E on H are weakened or even broken in abundant water periods. All feedback loops are strengthened in low flow periods accompanied by their greater or smaller values of $LRR_n$ than other periods. If there was only water donation, all values of $LRR_n$ for the reservoirs are lower than those without IWDPs, while all values of $LRR_n$ for reservoirs are greater than those without IWDPs. Water donation has negative impacts on the negative feedbacks between S and H, on the negative feedbacks between S and E, and on the positive feedbacks between H and E. While water receiving has positive impacts on these feedbacks. Less positive feedbacks are found with IWDPs than without them. Feedback loops of SHE nexus exhibit intrinsic similarity and stability across different time scales. The impact of reservoir operation and regulation on SHE nexus are clearer at the monthly scale. The seasonal scale offers the variations in periodic feedback loops. And the annual scale offers inter-annual and spatial trends of the SHE nexus from a macro perspective. Feedback loops in reservoirs with regulation function (e.g., AK and DJK) can remain stable under the varying inflow conditions at monthly scale. The positive feedbacks between H and E are weakened or even turn to be negative in the small installed hydropower generation capacity reservoirs (e.g., the XL reservoir) even in abundant water periods. Feedback loops for downstream reservoirs are influenced by their upstream reservoirs, especially in low flow periods.

This framework offers a systematic and quantitative approach to examining the spatiotemporal variations of SHE nexus with external perturbations. It elucidates the existence and nature of collaborative states among S, H, and E. However, more work should be done to enrich the representation of every component such as the E component. This component should be reflected by a comprehensive set of water quality indicators. Then more details of the mechanism of the SHE nexus will be figured out.

*Code and data availability.* The code and data that supports the findings of this study is available from the corresponding author upon reason able request.

*Declaration of competing interest.* The authors declare that they have no known competing financial interests or personal relationships that could have appeared to influence the work reported in this paper.

*Author contributions.* JW: Writing - original draft, Methodology, Investigation, Formal analysis, Data curation,



Conceptualization. DL: Conceptualization, Supervision, Project administration, Data curation, Funding acquisition, Writing – review & editing. SG, LX, HC, JC, and JY: Supervision, Project administration, Writing – review & editing. YZ: Methodology, Writing – review & editing.

*Competing interests*. The authors declare that they have no conflict of interest.

*Financial support.* This research has been supported by the National Natural Science Foundation of China (No. 52379022) and
the National Key Research and Development Project of China (2022YFC3202803).

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

7.