# Peer review of "Impacts of Inter-basin Water Diversion Projects on the Feedback Loops of Water Supply-Hydropower Generation-Environment Conservation Nexus"

_Hydrology and Earth System Sciences, 2024_

## Author Comment (AC1)

**Reference Number: hess-2024-399-RC1**

**RESPONSES TO REVIEWER ONE'S COMMENTS**

We would like to express our sincere appreciation for your professional and insightful remarks on our paper. The comments are valuable and helpful for us to improve the quality of the manuscript. All the concerns raised have been carefully treated and an itemized reply to the reviewer's comments is presented in the revision files.

**Major comments/questions**

**Point #1**

**COMMENT:** *Line 129 to 131: How do the dam release rules factor into the estimation of dam discharge, and are they integrated alongside the catchment area ratio?*

**RESPONSE:** The authors much appreciate the reviewer's insightful comments and apologetic for not clearing discussing the estimation of dam discharge. Lines 129-131 describe the method for calculating the runoff to the primary reservoir and the interval runoff of each pair reservoirs, but the formula for the inflow to the $i$th reservoir was not included. Specifically, the discharge from the reservoir is determined by the inflow and the specific operational rules of the reservoir. The inflow to the primary reservoir in a reservoirs group is calculated using the runoff from the hydrological stations simulated by the VIC model and the ratio of the catchment area. The inflow to the $i$th reservoir is the sum of the discharge from the $(i-1)$th reservoir and the interval runoff. The interval runoff for each reservoir is calculated using the runoff simulated by the VIC model and the catchment area ratio. The discharge for each reservoir is allocated according to its regular operational rules and the rules set for each scenario (flood control is the primary requirement, and the scheduling rules are adjusted according to different combinations of priorities for water supply, hydropower generation, and environment conservation). To address the issue of the missing explanation in the manuscript, modifications have been made in lines 129-131, and relevant references have been added. The revised and relevant parts are:

"After getting the acceptable runoff simulation results at the selected hydrological stations, the

runoff to reservoirs and the interval runoff of each pair reservoirs are estimated according to the catchment area ratio of each reservoir with its upstream and downstream hydrological stations. The calculation formulas are as follows:

$$
Q_{i,t}^{s} =
\begin{cases}
\dfrac{Q_{d,1,t}^{s} \times A_{1}}{A_{d,1}}, & i = 1 \\[4mm]
Q_{u,i,t}^{s} + \dfrac{\left(Q_{d,i,t}^{s} - Q_{u,i,t}^{s}\right) \times \left(A_{i} - A_{u,i}\right)}{\left(A_{d,i} - A_{i}\right)}, & i > 1
\end{cases}
\tag{4}
$$

$$
\Delta Q_{i,t} = Q_{i,t}^{s} - Q_{i-1,t}^{s}, \quad i > 1
\tag{5}
$$

where $Q_{i,t}^{s}$ is the runoff to the $i$th reservoir at $t$th period, m³/s; $Q_{u,i,t}^{s}$ and $Q_{d,i,t}^{s}$ are the simulation runoff results of the upstream and downstream hydrological stations of the $i$th reservoir at $t$th period, m³/s; $A_{i}$ is the catchment area of $i$th reservoir, m²; $A_{u,i}$ and $A_{d,i}$ are the catchment areas of the upstream and downstream hydrological stations, m². $\Delta Q_{i,t}$ is the interval runoff of the $i$th reservoir at $t$th period, m³/s.

The inflow to the $i$th reservoir is the sum of the discharge from the ($i$-1)th reservoir and the interval runoff. The calculation formulas are as follows:

$$
Q_{i,t} =
\begin{cases}
Q_{1,t}^{s}, & i = 1 \\[2mm]
Q_{out,i-1,t} + \Delta Q_{i,t}, & i > 1
\end{cases}
\tag{6}
$$

where $Q_{i,t}$ is the inflow to the $i$th reservoir at $t$th period, m³/s; $Q_{out,i-1,t}$ is the water release from the ($i$-1) th reservoir in period $t$, m³/s." (**On page 5-6 of the revised manuscript**)

**Point #2**

*COMMENT: Section 2.3 Was the FDC constructed using naturalised flows or the current/modified flows in this study? What are the implications?*

*RESPONSE:* We are very thankful for the reviewer's insightful comment and valuable reminder.

In this manuscript, the FDC was constructed using the simulated runoffs from 1976-2020 by VIC model. The FDC was constructed in order to find the discharges at the different percent duration points for various river sections, water years (e.g., wet, normal, and dry years), and months, and these discharges are taken as multi-level ecological flow standards. The MTMMHC method, combined with the modified FDC, can solve four key problems existed in the current ecological flow standards: spatial transferability, monthly variability, inter-annual variability and scalability (Li, et al., 2015). This method has been widely applied in various river basins (Li and Kang, 2014), with multiple simulations conducted for ecological flow standards and classification in the HRB. The results of this manuscript align well with these studies in terms of EF trends, flow ranges, and grading number across different water years and months, thus providing support and validation for our results (Li and Kang, 2014; Zhang and Liu, 2023). To clarify this point, relevant statements and references have been added in the revised manuscript:

> "The year groups are divided into wet years (precipitation below the 25th percentile, P<25 %), normal years (25 %≤P≤75 %), and dry years(P>75 %) firstly. Then, a flow duration curve (FDC, Franchini et al., 2011) is constructed using the total-period method based on daily average flows simulated from 1976-2020 by VIC model. Finally, the average of flows corresponding to the 90th and 95th percentiles of the FDC ($Q_{(90)xy}$ and $Q_{(95)xy}$, $m^3/s$) for the $y$th month of the $x$th year is taken as the Minimum Ecological Flow ($MEF_{xy}$, $m^3/s$)." (**On page 6 of the revised manuscript**)

**Relevant references:**

Li, C., Kang, L., Zhang S., Zhou, L.W.: A Modified FDC Method with Multi-level Ecological Flow Criteria, J. Yangtze River Sci. Res. Inst., 32 (11): 1-6, 13, https://doi.org/10.11988 /ckyyb.20140814, 2015. (in Chinese)

Li, C., and Kang, L.: A New Modified Tennant Method with Spatial-Temporal Variability, Water Resour. Manag., 28(14), 4911-4926, https://doi.org/10.1007/s11269-014-0746-4, 2014.

Zhang, X., and Liu, D.: A Method to determine the threshold of water exploitation index based on ecological flow estimation., China Rural Water and Hydropower, 2023 (2): 88-100+107. https://doi.org/10.12396/znsd.221653, 2023. (in Chinese)

**Point #3**

**COMMENT:** *Section 2.3 / Table 4: The MTMMHC method effectively sets ecological flows*

*retrospectively, but how can it be adapted for real-time dam operations? How can operational decisions account for the significant variation in MEF between wet and dry years, especially when such conditions are uncertain at the start of the year?*

**RESPONSE:** We greatly appreciate the reviewer's insightful comments and apologize for not clearly discussing the applicability of the MTMMHC method in real-time dam operations in the original manuscript. Ecological flow (EF) refers to the minimum flow required to sustain the health and function of aquatic ecosystems. There are over 200 methods for EF assessment (EFA) worldwide, typically categorized into four types: hydrological, hydraulic, habitat simulation, and holistic methods (Tharme, 2003). Traditionally, ecological flow is estimated using a percentage of the long-term average annual flow, without accounting for the effects of reservoir operations. The Tennant method, which determines EF based on predetermined percentages of average annual flow, is the most widely used hydrological method (Tharme, 2003). The MTMMHC method builds upon the Tennant method, modifying it based on three parameters: average periodic flow, water period, and percentage (Li and Kang, 2014). This modification helps mitigate the impacts of extreme inter-annual flow variations and uneven intra-annual distribution. In this study, a multi-level ecological flow standard is established through the MTMMHC method, which is determined by runoffs in various river sections, water years (e.g., wet, normal, and dry years), and months, and is independent of specific reservoir operations. All scenarios are modeled using the same ecological flow standard to clarify the differences in their environment conservation.

Accordingly, in the revised manuscript, we have added some content to express the applicability the MTMMHC method at the Methodology of the manuscript. The corresponding part is:

"In order to establish a multi-level ecological flow standard to aid in evaluating river ecological health, the multi-level ecological flows are estimate by the MTMMHC method. There are over 200 methods for EFs estimation worldwide, typically categorized into four types: hydrological, hydraulic, habitat simulation, and holistic methods (Tharme, 2003). The Tennant method, which determines EFs based on predetermined percentages of average annual flow, is the most widely used hydrological method (Tharme, 2003). The MTMMHC method (Li and Kang, 2014) modifies the Tennant method based on three parameters: average periodic flow, water period, and percentage. It

can solve four key problems existed in the current ecological flow standards: spatial transferability, monthly variability, inter-annual variability and scalability (Li, et al., 2015). Indeed, the MTMMHC method can avoid the impacts of extreme inter-annual flow events and uneven intra-annual distribution. This enables the calculation of different guarantee rates for various river sections, water years (e.g., wet, normal, and dry years), and months. It reflects the temporal and spatial variability of EFs, and provides a comprehensive and reasonable multi-level ecological flows standards. The steps of the MTMMHC method are as follows." (**On page 6 of the revised manuscript**)

**Relevant references:**

Tharme, E.: A global perspective on environmental flow assessment: emerging trends in the development and application of environmental flow methodologies for rivers. River Res. Appl. 19(5-6): 397–441, https://doi.org/10.1002/rra.736, 2003.

Li, C., and Kang, L.: A New Modified Tennant Method with Spatial-Temporal Variability, Water Resour. Manag., 28(14), 4911-4926, https://doi.org/10.1007/s11269-014-0746-4, 2014.

**Point #4**

**COMMENT:** *Figures (from figure 7): The caption for the figures should be more informative. For Figures 7 and 8, for instance, it should state the scenarios, with or without IWDP respectively, the priorities and what each LRR represents (S, H or E). Also, it would be good for Figure 8 to be immediate below 7 so readers can easily compare the effect if IWDPs.*

**RESPONSE:** The authors are very thankful for the reviewer's insightful comments and helpful suggestions. We have added this information to the caption of Figures: state the scenarios, with or without IWDP respectively, the priorities, and what each *LRR* represents (S, H, or E), and we have listed Figures 7 and 8 together. The revised parts are:

[revised manuscript text omitted]

**Point #5**

***COMMENT:*** *Line 454- 464: What metrics were used to quantify runoff variations across time scales? Was the link between runoff and feedback loops validated?*

***RESPONSE:*** We greatly appreciate the reviewer's insightful comments and their thorough thinking and guidance on this study. We apologize for not providing a more in-depth discussion and explanation of this issue in the manuscript, which led to your confusion. To verify the results, wavelet transform analysis of runoff for HJX, AK, DJK, WFZ, and XL dam sites, as shown in

Fig. 1. It can be found that the runoff in all reservoirs exhibits strong periodicity at a time scale of 4-8 months during 2006-2017, while downstream reservoirs (i.e., DJK, WFZ, and XL) show strong periodicity at 1-3 months during 2018-2020. Overall, the runoff exhibits stronger periodicity at the 3-month scale, which provides strong evidence that the seasonal results can help analyze the variations in periodic feedback loops. The link between runoff and feedback loops is determined by comparing the values of $LRR_n$ with the runoff at different spatiotemporal scales. We found that the trends in $LRR_n$ and runoff show similar patterns in their spatiotemporal evolution, and the mathematical implications of $LRR_1$, $LRR_2$, and $LRR_3$ (e.g., differences in water supply, hydropower generation, and ecological flow satisfaction rates under different scenarios) suggest that runoffs are the key factors determining the $LRR_n$ values. To make it easier to understand, we give an example here: Fig. 2 illustrates $LRR_1$ (i.e., the log response ratio of the S component) between $S_{2\text{-}3\text{-}2\text{-}1}$ and $S_{2\text{-}3\text{-}4\text{-}1}$, $LRR_2$ (i.e., the log response ratio of the H component) between $S_{2\text{-}3\text{-}1\text{-}2}$ and $S_{2\text{-}3\text{-}4\text{-}2}$, $LRR_3$ (i.e., the log response ratio of the E component) between $S_{2\text{-}3\text{-}1\text{-}3}$ and $S_{2\text{-}3\text{-}4\text{-}3}$ and runoff for HJX dam sites. We also conducted a Granger causality test between $LRR_n$ and runoffs and found significant causal links. However, since this part is not the focus of this study, in the revised version, we have enriched the presentation, but no longer present the results of wavelet transform analysis of runoff and the Granger causality test between $LRR_n$ and runoffs.

[Figure]

[Figure]

Fig.1. Wavelet transform analysis of runoff for HJX, AK, DJK, WFZ, and XL dam sites.

[Figure]

[Figure]

Fig. 2. *LRR*$_1$ (i.e., the log response ratio of the S component) between S$_{2-3-2-1}$ and S$_{2-3-4-1}$, *LRR*$_2$ (i.e., the log response ratio of the H component) between S$_{2-3-1-2}$ and S$_{2-3-4-2}$, *LRR*$_3$ (i.e., the log response ratio of the E component) between S$_{2-3-1-3}$ and S$_{2-3-4-3}$ and runoffs for HJX dam site.

The corresponding part is:

"Therefore, signs of mean values of *LRR*$_n$ at seasonal and annual scales are consistent with those at monthly scale, so the feedback loops of SHE nexus exhibit intrinsic similarity and stability across different time scales. Compared with the values of *LRR*$_n$ at monthly scale, the values at the seasonal scale show its stronger periodic variations. Based on the variations in *LRR*$_n$ and the mathematical implications of *LRR*$_1$, *LRR*$_2$, and *LRR*$_3$, this study found that these periodic variations align closely with the runoff variations, and the temporal and spatial variations in feedback loops are primarily attributed to variations in runoff. The wavelet transform analysis has also been applied in the runoffs for HJX, AK, DJK, WFZ, and XL dam sites. And the results are in consisted with that in Hutuo River Basin (Xu et al., 2018), the periodic variations have been found at the seasonal scale. The *LRR*$_n$ values at the seasonal scale can help analyze the variations in periodic feedback loops. Different from the monthly or seasonal scales, results at the annual scale reveal the long-term trends and periodic variations in the inter-annual and spatial trends of the SHE nexus from a macro perspective. The impacts of reservoir operation and the regulation on SHE nexus can be clearly simulated and observed at the monthly scale, so the immediate changes in the nexus at monthly scale can provide information for short-term decision-making in reservoirs." (**On page 20-21 of the revised manuscript**)

**Point #6**

*COMMENT: Results and Discussion: Very little discussion or reference to other studies. For instance, no comparison to real world observations from the HRB; have any of the scenarios occurred in reality? And if so, were the feedback loops in line with the findings? Also, the impacts of IWDPs on feedback loops are reported, but how do these findings translate into actionable management strategies? Are there optimal thresholds for water donation and receiving that maximize system-wide stability of the SHE nexus? How can this framework guide policy or reservoir operation strategies in basins like HRB? Are there specific recommendations for balancing S, H and E, especially in low flow months, where competition between water supply, hydropower, and environmental needs intensifies?*

*RESPONSE:* We much appreciate the reviewer's insightful comments and apologetic for the lack of discussion or reference with other studies in the original manuscript. In the revised manuscript, we have added more discussions based on real world observations from the HRB and relevant studies. In addition, for the Han-to-Wei Water Diversion Project (Wei et al., 2020), the Middle Route of the South-to-North Water Diversion Project (Li et al., 2016), the Northern Hubei Water Resources Allocation Project (He and X, 2020), and the Changjiang-to-Han River Water Diversion Project (Zhang et al., 2022) discussed in Sections 4.3 and 4.4, the actual (trial) water diversion times are as follows: 2023, 2014, 2021, and 2014, respectively. The Three Gorges Reservoir to Hanjiang River (Yang et al., 2012) is still under construction and has not yet been diverted, so long-term research based on real-world conditions cannot be conducted. Therefore, this manuscript constructs a Multisource Input-Output Reservoir Generalization (MIORG) model based on the operational conditions of IWDPs, reservoir parameters, and scheduling rules, with long-term scale runoff inputs, to address the different impacts of IWDPs on the dynamic SHE nexus with multiple scenarios. Thus, in the Results and Discussion section, we have added discussions between the relevant studies in HRB and our results.

Based on the results from this manuscript, we have found that water donation has negative impacts on the negative feedbacks between S and H, on the negative feedbacks between S and E, and on the positive feedbacks between H and E. While water receiving has positive impacts on these feedbacks. Additionally, upstream IWDPs have a significant influence on the downstream

SHE nexus. In our future research, a model will be developed to simulate SHE nexus system, and the optimal thresholds for water donation, water receiving and water resource utilization will be determined through optimal algorithms and deep learning models.

Regarding the results in this study, we can provide some recommendations: water donation or regional water supply can be increasing in abundant water periods in order to reduce spilled water and increase hydropower generation efficiency. In dry periods, it is necessary to consider the priority order of the water supply, hydropower generation, environment conservation, determine water utilization threshold for each component to maximize the benefits. We have added several water management recommendations to the conclusion.

We have made extensive revisions in the manuscript:

[revised manuscript text omitted]

**Minor comments**

**Point #7**

**COMMENT:** *Line 107: "It has been widely application". Correct to "It has been widely*

*applied".*

*RESPONSE:* We much appreciate and totally agree with the reviewer's insightful comment. The revised part is:

"It has been widely applied in runoff simulations across various basins worldwide, consistently yielding outstanding results."

**Point #8**

*COMMENT: Line 125: "approaching 1 meant". Correct to "approaching 1 means".*

*RESPONSE:* The authors much appreciate your thoughtful comment. We agree with the reviewer's point and will revise the sentence accordingly for clarity. The revised part is:

"$R^2$ approaching 1 means the simulations are equal to the observations."

**Point #9**

*COMMENT: Line 145: You need to state that P is precipitation (I assume P<25% means precipitation below the 25th percentile).*

*RESPONSE:* We are very thankful for the reviewer's helpful suggestions and apologetic for providing an improper description in the original manuscript. In the revised manuscript, we have accordingly modified the description to clarify it more accurately and enhance the rigor of the article. The revised part is:

"① The year groups are divided into wet years (precipitation below the 25th percentile, P<25 %), normal years (25 %≤P≤75 %), and dry years(P>75 %) firstly. (**On page 6 of the revised manuscript**)"

**Point #10**

*COMMENT:* *Figure 3: The arrows of outflows (reg. water supply flow, ET and seepage, water donation) start at different locations for the ith reservoir and the (i+1) th reservoir.*

*RESPONSE:* The authors much appreciate the reviewer's insightful comment and apologetic for not proving typical references to support this statement. The revised parts are:

"

**Figure 3. The multisource input-output to reservoirs in a reservoirs group."**

**Point #11**

*COMMENT:* *Line 229: Should this read: "Thus, the differences between Nexus I and Nexus III can figure out impact of different IWDP clusters on the SHE nexus"?*

*RESPONSE:* We sincerely appreciate the reviewer's thoughtful comment and constructive suggestion. We agree with the reviewer's point and will revise the sentence accordingly for clarity. The revised part is:

"Thus, the differences between Nexus I and Nexus II can figure out the impacts of IWDPs on the

SHE nexus. To identify the SHE nexus with different clusters of IWDPs (i.e., the feedback loops of Nexus III as shown in Figure 1.), the differences between $S_{2\text{-m-p-c}}$ and $S_{1\text{-0-4-c}}$ scenarios are determined. Thus, the differences between Nexus I and Nexus III can figure out the impacts of different IWDP clusters on the SHE nexus. $S_{1\text{-0-4-c}}$ and $S_{2\text{-3-4-c}}$, are the baseline scenarios for distinguishing Nexus I, Nexus III, and Nexus II. In the same way, to clarify the impacts of IWDPs on the three components, the differences between the $S_{1\text{-0-4-c}}$ and $S_{2\text{-3-4-c}}$ scenarios are determined. (**On page 10 of the revised manuscript**)"

**Point #12**

*COMMENT: Table 4: What are the units of the e-flows?*

*RESPONSE:* We are very thankful for the reviewer's insightful comments and helpful suggestions. The units of the e-flows are $m^3/s$, the additions are made in Table 4 as follows:

**Table 4**. Multi-level ecological flows resulted from MTMMHC method.

| Site | Month | Wet year | | | | Normal year | | | | Dry year | | | |
|---|---|---|---|---|---|---|---|---|---|---|---|---|---|
| | | *MEF* ($m^3/s$) | $E_2$ ($m^3/s$) | $OEF_{min}$ ($m^3/s$) | $OEF_{max}$ ($m^3/s$) | *MEF* ($m^3/s$) | $E_2$ ($m^3/s$) | $OEF_{min}$ ($m^3/s$) | $OEF_{max}$ ($m^3/s$) | *MEF* ($m^3/s$) | $E_2$ ($m^3/s$) | $OEF_{min}$ ($m^3/s$) | $OEF_{max}$ ($m^3/s$) |
| XL dam site | Jan | 1197 | 1476 | 1550 | 1668 | 825 | 849 | 872 | 910 | 664 | 666 | 668 | 670 |
| | Feb | 1265 | 1467 | 1539 | 1656 | 836 | 863 | 890 | 933 | 675 | 678 | 681 | 686 |
| | Mar | 1268 | 1486 | 1569 | 1702 | 842 | 869 | 896 | 938 | 685 | 690 | 696 | 705 |
| | Apr | 1249 | 1329 | 1426 | 1581 | 868 | 892 | 916 | 955 | 691 | 698 | 704 | 714 |
| | May | 1273 | 1675 | 1822 | 2058 | 861 | 887 | 912 | 953 | 705 | 714 | 723 | 738 |
| | Jun | 1653 | 1681 | 1877 | 2192 | 877 | 916 | 955 | 1017 | 763 | 786 | 809 | 846 |
| | Jul | 1818 | 2629 | 2987 | 3560 | 1288 | 1430 | 1572 | 1799 | 875 | 921 | 968 | 1043 |
| | Aug | 1885 | 2522 | 2849 | 3372 | 1266 | 1401 | 1537 | 1753 | 811 | 845 | 879 | 933 |
| | Sep | 1465 | 2822 | 3225 | 3869 | 1174 | 1279 | 1384 | 1553 | 834 | 879 | 924 | 997 |
| | Oct | 1368 | 2276 | 2611 | 3148 | 978 | 1036 | 1094 | 1186 | 733 | 752 | 772 | 802 |
| | Nov | 1315 | 1586 | 1748 | 2007 | 897 | 932 | 966 | 1022 | 691 | 697 | 704 | 714 |
| | Dec | 1194 | 1471 | 1549 | 1675 | 845 | 873 | 900 | 944 | 680 | 686 | 691 | 700 |

Generally, we are deeply grateful to the reviewer #1 for his/her insightful and careful review. The provided comments and suggestions have greatly helped improve the manuscript. We also expressed our gratitude in the "**Acknowledgments**" section of the revised manuscript.

---

## Author Comment (AC2)

**Reference Number: hess-2024-399-RC2**

**RESPONSES TO REVIEWER ONE'S COMMENTS**

We would like to express our sincere gratitude for your detailed and constructive comments on our manuscript. The comments are valuable and helpful for us to improve the quality of the manuscript. All the concerns raised have been carefully treated and an itemized reply to the reviewer's comments is presented in the revision files.

**Point #1**

**COMMENT:** *While the methodology employed in the manuscript effectively addresses the issue of identifying the SHE nexus across multiple temporal and spatial scales, it is important to elaborate on the advantages of the chosen approach in the methodology or introduction section, rather than merely stating its ability to solve the problem.*

**RESPONSE:** We sincerely appreciate the reviewer's valuable comment regarding the need to elaborate on methodological advantages. We fully concur that explicitly articulating the strengths of our chosen analytical framework in the methodology/introduction sections will better contextualize our approach for readers. In the revised manuscript, we will expand upon some key advantages of our methodology in addressing the impacts of IWDPs across the multiple temporal and spatial scales on the dynamic SHE nexus. The Variable Infiltration Capacity (VIC) hydrological model offers significant advantages in multiple temporal and spatial scale runoff simulation. It has flexible spatial resolution, making it suitable for hydrological modeling at scales ranging from small catchments to large basins, with minimal loss of accuracy. VIC model can simulate hydrological processes at various time scales, from hourly to annual, catering to different research needs. The VIC model also efficiently uses gridded data, making it highly adaptable for large-scale regional or global studies, and supports a wide range of input data types. The Modified Tennant Method Based on Multilevel Habitat Conditions method builds upon the Tennant method, modifying it based on three parameters: average periodic flow, water period, and percentage (Li and Kang, 2014). It can solve four key problems existed in the current ecological flow standards: spatial transferability, monthly variability, inter-annual variability and scalability (Li, et al., 2015). This modification helps mitigate the impacts of extreme inter-annual

flow variations and uneven intra-annual distribution. The Log Response Ratio method captures non-linear feedback loops within complex SHE nexus systems. And our scenarios architecture enables systematic exploration of SHE nexus systems by combining different clusters of IWDPs and the priority orders of S, H, and E, offering flexibility in modeling system behavior under different conditions.

The revised and relevant parts are:

"To simulate runoff results at multiple temporal and spatial scales, the Variable Infiltration Capacity (VIC) hydrological model is selected. The Variable Infiltration Capacity (VIC) hydrological model offers significant advantages in multiple temporal and spatial scale runoff simulation. It is a large-scale distributed hydrological model based on the spatial distribution grid of Soil Vegetation Atmospheric Transfer Schemes (SVATS) (Liang, et al., 1994), making it highly adaptable to studies at different spatial scales and supporting a wide range of input data types. The VIC model can simulate hydrological processes at various time scales, from hourly to annual, catering to different research needs. It excelled at simulating both the energy balance and water balance between the land and atmosphere, thereby addressing the oversight of energy processes in traditional hydrological models. The VIC model has been widely applied in runoff simulations across various basins worldwide, consistently yielding outstanding results. There are five steps to construct a VIC model (Koohi et al., 2022): ① collect and organize data; ② preprocesses of the VIC model; ③ construct VIC model of the selected basin; ④ run the catchment module; ⑤ parameter calibration and validation. During the calibration process, important parameters highlighted in Table 1 are automatically calibrated using MATLAB to achieve the optimal parameter combination."(**On page 4 of the revised manuscript**)

"In order to establish a multi-level ecological flow standard to aid in evaluating river ecological health, the multi-level ecological flows are estimate by the MTMMHC method. There are over 200 methods for EFs estimation worldwide, typically categorized into four types: hydrological, hydraulic, habitat simulation, and holistic methods (Tharme, 2003). The Tennant method, which determines EFs based on predetermined percentages of average annual flow, is the most widely used

hydrological method (Tharme, 2003). The MTMMHC method (Li and Kang, 2014) modifies the Tennant method based on three parameters: average periodic flow, water period, and percentage. It can solve four key problems existed in the current ecological flow standards: spatial transferability, monthly variability, inter-annual variability and scalability (Li, et al., 2015). Indeed, the MTMMHC method can avoid the impacts of extreme inter-annual flow events and uneven intra-annual distribution. This enables the calculation of different guarantee rates for various river sections, water years (e.g., wet, normal, and dry years), and months. It reflects the temporal and spatial variability of EFs, and provides a comprehensive and reasonable multi-level ecological flows standards. The steps of the MTMMHC method are as follows." (**On page 6 of the revised manuscript**)

"To analyse the feedback loops in Nexus I, Nexus II and Nexus III in Figure 1, the log response ratio (*LRR*) quantization method (Patrick et al., 2022) is used to quantify the responses of S, H, and E with different clusters of IWDPs. This method captures non-linear feedback loops within complex SHE nexus systems." (**On page 9 of the revised manuscript**)

"To identify the impacts of different clusters of IWDPs on the SHE nexus, scenarios are set according to the following three aspects: with or without IWDPs (i.e., two types for IWDPs), different clusters of IWDPs (i.e., four clusters for the above two types), and the priority orders of S, H, and E. As there are three components for the highest priority, six scenarios can be obtained through the combination of the three components. As all S, H, and E are determined from standard scheduling rules, there are also three types for the standard scheduling rules. Combined with the types of different clusters of IWDPs, there will be a total of 30 scenarios (i.e., 4 clusters of IWDPs ×6 types for the highest priority combinations +2 types for IWDPs ×3 types for standard scheduling rules) as listed in Table 2. Specifically, to iteratively set the priority orders of S, H, and E, all three components are all in standard scheduling rules firstly. Secondly, the highest priority is set to water supply (as denoted by S-Priority), with the regional water supply increased to 120 %. And thirdly, hydropower generation (H-Priority) is prioritized to achieve the maximum output during the planned period. Finally, environmental conservation (E-Priority) is addressed through ensuring that the reservoir outflow meets $OEF_{xy(\max)}$. These scenarios offer flexibility in modeling SHE nexus

system behavior under different conditions." (**On page 10 of the revised manuscript**)

**Point #2**

*COMMENT: The elements presented in Figure 4 are insufficient to clearly illustrate the geographical characteristics of the study area. Additionally, it is necessary to label the names of various hydrological stations and reservoirs on the map, so that readers can more easily interpret the information. The clarity of Figure 6 should be improved, and the color scheme used to differentiate observed and simulated data needs to be adjusted for better distinction. Furthermore, the title of Figure 6 could be simplified for conciseness.*

*RESPONSE:* We are very thankful for the reviewer's insightful comment and valuable reminder. We have revised Figure 4 to enhance the geographical characteristics of Hanjiang River Basin (HRB) by adding elements such as topography and rivers to make the map clearer. We have also labeled the hydrological stations and reservoirs on the map, ensuring that readers can easily identify these key locations. To eliminate readers' disputes over the territories in the map, we have made modifications to Figure 4 using the map with the examination approval number GS (2024) No.0650. Regarding Figure 6, we have improved its clarity by ensuring that text, line thickness, and other elements are sharp and legible. Additionally, we have adjusted the color scheme used to differentiate observed and simulated data, opting for more contrasting colors that are easily distinguishable, and have ensured the legend clearly indicates which color corresponds to each dataset. Lastly, we have simplified the title of Figure 6 to a more concise. The revised and relevant parts are:

[Figure]

**Figure 4. Overview map of the study area.**

[Figure]

Figure 6. Calibration and validation results of simulation at hydrological stations: (a)Xiangjiangping, (b) Baihe, (c) Huanglongtan, (d) Huangjiagang, (e) Xiangyang, (f) Huangzhuang."

**Point #3**

*COMMENT: Is the framework proposed in the manuscript broadly applicable? It might be helpful for the manuscript to provide a clearer explanation of the framework and further clarify*

*the scope of applicability of the proposed method.*

***RESPONSE:*** We greatly appreciate the reviewer's insightful comments. This framework offers a systematic and quantitative approach to examining the spatiotemporal variations of SHE nexus with external perturbations. It elucidates the existence and nature of collaborative states among S, H, and E. All the methods in the framework, such as the VIC model, the Modified Tennant Method Based on Multilevel Habitat Conditions, and the Log Response Ratio method, are not region-specific and can be applied to the study of SHE nexus in different basins worldwide. Therefore, the proposed framework can be applied globally to identify the feedbacks of the SHE nexus in basins with inter-basin water diversion projects. The applicability of the framework is clearly explained in the paper. The corresponding part is:

> "To address the impacts of IWDPs across the multiple temporal and spatial scales on the dynamic SHE nexus, multiple temporal and spatial scales runoffs from the water donating basins are provided through a distributed hydrological model. And multi-level ecological flows and their corresponding multi-level ecological flow standards are also determined according to an available method with spatial-temporal variability. To facilitate the identification of the impacts of IWDPs on SHE nexus, scenario experiments are set by "with/without IWDPs". In order to take the different clusters of IWDPs into account, scenario experiments are classified by the impacts of IWDPs on water donation area, on water receiving area or on an area with both water donation and water receiving if there are IWDPs. To evaluate the feedback loops of the SHE nexus, the priority order of S, H, and E are iteratively set in all reservoir nodes. We set different types of the highest priority in S, H, and E (i.e., S-Priority, H-Priority, and E-Priority) and take the standard scheduling rules as reference scenarios. All scenarios are modeled in a multisource input-output reservoir generalization model, and differences between scenarios are quantified with a response ratio indicator. And the feedback loops with the different impacts of IWDPs are identified through a response ratio indicator. To explore the collaborative states, positive mutation in a response ratio across time-space is found between pairwise components of SHE. This framework can be applied globally to identify the feedbacks of the SHE nexus in basins with IWDPs. Thus, our research framework is illustrated as Figure 1." (**On page 3 of the revised manuscript**)

**Point #4**

*COMMENT: The manuscript offers limited description of the baseline scenarios. This section could be expanded to clarify the rationale behind the selection of the baseline scenarios, enabling readers to better understand the results.*

*RESPONSE:* We are very thankful for the reviewer's insightful comments and helpful suggestions. We have provided a more detailed description of the baseline scenarios and added explanations of the scenarios in the figure captions.

The revised parts are:

[revised manuscript text omitted]

**Point #5**

*COMMENT: The results and discussion section is too long, please make it more concise and highlight the key results.*

*RESPONSE:* We agree that the section could be more concise and focused on the key findings. In response to this comment, we have increased the analysis of the results and reduced the repetition of the results in the charts. And we have refined the analysis and discussion of similar

results at different scales, remove repetitive expressions, and emphasize the differences in results caused by different spatial and temporal scales. We believe these changes have made the Results and Discussion section more concise while maintaining the scientific rigor and clarity of our findings. The revised parts are:

[revised manuscript text omitted]

Generally, we are deeply grateful to the reviewer #2 for his/her insightful and careful review. The provided comments and suggestions have greatly helped improve the manuscript. We also expressed our gratitude in the "**Acknowledgments**" section of the revised manuscript.

---

## Author Response (AR1)

**Cover letter**

April 3rd, 2025

**Number: hess-2024-399**

**Manuscript Title:** Impacts of Inter-basin Water Diversion Projects on the Feedback Loops of Water Supply-Hydropower Generation-Environment Conservation Nexus

Dear Prof. Pieter van der Zaag,

We would like to express our sincere gratitude for the opportunity to submit our manuscript entitled "Impacts of Inter-basin Water Diversion Projects on the Feedback Loops of Water Supply-Hydropower Generation-Environment Conservation Nexus" to Hydrology and Earth System Sciences. We much appreciate your professional and insightful comments. All the concerns raised have been carefully treated and an itemized reply to your comments is presented in the revision files. Our changes are marked in Red in the revised manuscript.

Thank you very much again for your time and kind help. Looking forward to hearing from you.

Sincerely yours,

Dedi Liu Email: dediliu@whu.edu.cn

**RESPONSES TO REVIEWER #1'S COMMENTS**

We would like to express our sincere appreciation for your professional and insightful remarks on our paper. The comments are valuable and helpful for us to improve the quality of the manuscript. All the concerns raised have been carefully treated and an itemized reply to the reviewer's comments is presented in the revision files.

**Major comments/questions**

**Point #1**

**COMMENT:** Line 129 to 131: How do the dam release rules factor into the estimation of dam discharge, and are they integrated alongside the catchment area ratio?

**RESPONSE:** The authors much appreciate the reviewer's insightful comments and apologetic for not clearing discussing the estimation of dam discharge. Lines 129-131 describe the method for calculating the runoff to the primary reservoir and the interval runoff of each pair reservoirs, but the formula for the inflow to the *i*th reservoir was not included. Specifically, the discharge from the reservoir is determined by the inflow and the specific operational rules of the reservoir. The inflow to the primary reservoir in a reservoirs group is calculated using the runoff from the hydrological stations simulated by the VIC model and the ratio of the catchment area. The inflow to the *i*th reservoir is calculated using the runoff. The interval runoff for each reservoir is calculated using the runoff simulated by the VIC model and the catchment area ratio. The discharge for each reservoir is allocated according to its regular operational rules and the rules set for each scenario (flood control is the primary requirement, and the scheduling rules are adjusted according to different combinations of priorities for water supply, hydropower generation, and environment conservation). To address the issue of the missing explanation in the manuscript, modifications have been made in lines 129-131, and relevant references have been added. The revised and relevant parts are:

"After getting the acceptable runoff simulation results at the selected hydrological stations, the runoff to reservoirs and the interval runoff of each pair reservoirs are estimated according to the

catchment area ratio of each reservoir with its upstream and downstream hydrological stations. The calculation formulas are as follows:

$$Q_{i,t}^{s} = \begin{cases} \frac{Q_{d,l,t}^{s} \times A_{l}}{A_{d,l}}, i = 1\\ Q_{u,i,t}^{s} + \frac{\left(Q_{d,i,t}^{s} - Q_{u,i,t}^{s}\right) \times \left(A_{i} - A_{u,i}\right)}{\left(A_{d,i} - A_{i}\right)}, i > 1 \end{cases}$$
(4)

$$\Delta Q_{i,t} = Q_{i,t}^{s} - Q_{i-1,t}^{s}, i > 1$$
(5)

where  $Q_{i,t}^{s}$  is the runoff to the *i*th reservoir at *t*th period, m3/s;  $Q_{u,i,t}^{s}$  and  $Q_{d,i,t}^{s}$  are the simulation runoff results of the upstream and downstream hydrological stations of the *i*th reservoir at *t*th period, m3/s;  $A_{i}$  is the catchment area of *i*th reservoir, m2;  $A_{u,i}$  and  $A_{d,i}$  are the catchment areas of the upstream and downstream hydrological stations, m2.  $\Delta Q_{i,t}$  is the interval runoff of the *i*th reservoir at *t*th period, m3/s.

The inflow to the *i*th reservoir is the sum of the discharge from the (i-1)th reservoir and the interval runoff. The calculation formulas are as follows:

$$Q_{i,t} = \begin{cases} Q_{i,t}^{s}, i = 1\\ Q_{\text{out},i-1,t} + \Delta Q_{i,t}, i > 1 \end{cases}$$
(6)

where  $Q_{i,t}$  is the inflow to the *i*th reservoir at *t*th period, m3/s;  $Q_{out,i-1,t}$  is the water release from

the (*i*-1) th reservoir in period t, m3/s." (On page 5-6 of the revised manuscript)

**Point #2**

**COMMENT:** Section 2.3 Was the FDC constructed using naturalised flows or the current/modified flows in this study? What are the implications?

**RESPONSE:** We are very thankful for the reviewer's insightful comment and valuable reminder. In this manuscript, the FDC was constructed using the simulated runoffs from 1976-2020 by VIC model. The FDC was constructed in order to find the discharges at the different percent duration points for various river sections, water years (e.g., wet, normal, and dry years), and months, and these discharges are taken as multi-level ecological flow standards. The MTMMHC method, combined with the modified FDC, can solve four key problems existed in the current ecological flow standards: spatial transferability, monthly variability, inter-annual variability and scalability (Li, et al., 2015). This method has been widely applied in various river basins (Li and Kang, 2014), with multiple simulations conducted for ecological flow standards and classification in the HRB. The results of this manuscript align well with these studies in terms of EF trends, flow ranges, and grading number across different water years and months, thus providing support and validation for our results (Li and Kang, 2014; Zhang and Liu, 2023). To clarify this point, relevant statements and references have been added in the revised manuscript:

"The year groups are divided into wet years (precipitation below the 25th percentile, P<25 %), normal years (25 % $\leq$ P $\leq$ 75 %), and dry years(P>75 %) firstly. Then, a flow duration curve (FDC, Franchini et al., 2011) is constructed using the total-period method based on daily average flows simulated from 1976-2020 by VIC model. Finally, the average of flows corresponding to the 90th and 95th percentiles of the FDC ( $Q_{(90)xy}$  and  $Q_{(95)xy}$ , m3/s) for the *y*th month of the *x*th year is taken as the Minimum Ecological Flow ( $MEF_{xy}$ , m3/s)." (**On page 6 of the revised manuscript**)

**Relevant references:**

Li, C., Kang, L., Zhang S., Zhou, L.W.: A Modified FDC Method with Multi-level Ecological Flow Criteria, J. Yangtze River Sci. Res. Inst., 32 (11): 1-6, 13, https://doi.org/10.11988 /ckyyb.20140814, 2015. (in Chinese)

Li, C., and Kang, L.: A New Modified Tennant Method with Spatial-Temporal Variability, Water Resour. Manag., 28(14), 4911-4926, https://doi.org/10.1007/s11269-014-0746-4, 2014.

Zhang, X., and Liu, D.: A Method to determine the threshold of water exploitation index based on ecological flow estimation., China Rural Water and Hydropower, 2023 (2): 88-100+107. https://doi.org/10.12396/znsd.221653, 2023. (in Chinese)

**Point #3**

**COMMENT:** Section 2.3 / Table 4: The MTMMHC method effectively sets ecological flows retrospectively, but how can it be adapted for real-time dam operations? How can operational decisions account for the significant variation in MEF between wet and dry years, especially when such conditions are uncertain at the start of the year?

**RESPONSE:** We greatly appreciate the reviewer's insightful comments and apologize for not

clearly discussing the applicability of the MTMMHC method in real-time dam operations in the original manuscript. Ecological flow (EF) refers to the minimum flow required to sustain the health and function of aquatic ecosystems. There are over 200 methods for EF assessment (EFA) worldwide, typically categorized into four types: hydrological, hydraulic, habitat simulation, and holistic methods (Tharme, 2003). Traditionally, ecological flow is estimated using a percentage of the long-term average annual flow, without accounting for the effects of reservoir operations. The Tennant method, which determines EF based on predetermined percentages of average annual flow, is the most widely used hydrological method (Tharme, 2003). The MTMMHC method builds upon the Tennant method, modifying it based on three parameters: average periodic flow, water period, and percentage (Li and Kang, 2014). This modification helps mitigate the impacts of extreme inter-annual flow variations and uneven intra-annual distribution. In this study, a multi-level ecological flow standard is established through the MTMMHC method, which is determined by runoffs in various river sections, water years (e.g., wet, normal, and dry years), and months, and is independent of specific reservoir operations. All scenarios are modeled using the same ecological flow standard to clarify the differences in their environment conservation.

Accordingly, in the revised manuscript, we have added some content to express the applicability the MTMMHC method at the Methodology of the manuscript. The corresponding part is:

"In order to establish a multi-level ecological flow standard to aid in evaluating river ecological health, the multi-level ecological flows are estimate by the MTMMHC method. There are over 200 methods for ecological flows (EFs) estimation worldwide, typically categorized into four types: hydrological, hydraulic, habitat simulation, and holistic methods (Tharme, 2003). The Tennant method, which determines EFs based on predetermined percentages of average annual flow, is the most widely used hydrological method (Tharme, 2003). The MTMMHC method (Li and Kang, 2014) modifies the Tennant method based on three parameters: average periodic flow, water period, and percentage. It can solve four key problems existed in the current ecological flow standards: spatial transferability, monthly variability, inter-annual variability and scalability (Li, et al., 2015). Indeed, the MTMMHC method can avoid the impacts of extreme inter-annual flow events and uneven intra-annual distribution. This enables the calculation of different guarantee rates for various river sections, water years (e.g., wet, normal, and dry years), and months. It reflects the temporal and spatial variability of EFs, and provides a comprehensive and reasonable multi-level ecological flows standards. The steps of the MTMMHC method are as follows." (**On page 6 of the revised manuscript**)

**Relevant references:**

Tharme, E.: A global perspective on environmental flow assessment: emerging trends in the development and application of environmental flow methodologies for rivers. River Res. Appl. 19(5-6): 397–441, https://doi.org/10.1002/rra.736, 2003.

Li, C., and Kang, L.: A New Modified Tennant Method with Spatial-Temporal Variability, Water Resour. Manag., 28(14), 4911-4926, https://doi.org/10.1007/s11269-014-0746-4, 2014.

**Point #4**

**COMMENT:** Figures (from figure 7): The caption for the figures should be more informative. For Figures 7 and 8, for instance, it should state the scenarios, with or without IWDP respectively, the priorities and what each LRR represents (S, H or E). Also, it would be good for Figure 8 to be immediate below 7 so readers can easily compare the effect if IWDPs.

**RESPONSE:** The authors are very thankful for the reviewer's insightful comments and helpful suggestions. We have added this information to the caption of Figures: state the scenarios, with or without IWDP respectively, the priorities, and what each *LRR* represents (S, H, or E), and we have listed Figures 7 and 8 together. The revised parts are:

Figure 7. the differences of indexes (i.e.,  $LRR_1$ ,  $LRR_2$ ,  $LRR_3$  for log response ratio of the S, H, and E component) without IWDPs (i.e., between  $S_{0-p-n}$  and  $S_{0-4-n}$ ) at the monthly scale: (a-1) is  $LRR_2$  with the highest priority in S (i.e., between  $S_{0-1-2}$  and  $S_{0-4-2}$ ), (a-2) is  $LRR_3$  with the highest priority in S (i.e., between  $S_{0-1-3}$  and  $S_{0-4-3}$ ), (b-1) is  $LRR_1$  with the highest priority in H (i.e., between  $S_{0-2-1}$  and  $S_{0-4-1}$ ), (b-2) is  $LRR_3$  with the highest priority in H (i.e., between  $S_{0-2-3}$  and  $S_{0-4-3}$ ), (c-1) is  $LRR_1$  with the highest priority in H (i.e., between  $S_{0-2-3}$  and  $S_{0-4-3}$ ), (c-1) is  $LRR_1$  with the highest priority in H (i.e., between  $S_{0-2-3}$  and  $S_{0-4-3}$ ), (c-1) is  $LRR_1$  with the highest priority in H (i.e., between  $S_{0-2-3}$  and  $S_{0-4-3}$ ), (c-1) is  $LRR_1$  with the highest priority in H (i.e., between  $S_{0-2-3}$  and  $S_{0-4-3}$ ), (c-1) is  $LRR_1$  with the highest priority in H (i.e., between  $S_{0-2-3}$  and  $S_{0-4-3}$ ), (c-1) is  $LRR_1$  with the highest priority in H (i.e., between  $S_{0-2-3}$  and  $S_{0-4-3}$ ), (c-1) is  $LRR_1$  with the highest priority in H (i.e., between  $S_{0-2-3}$  and  $S_{0-4-3}$ ), (c-1) is  $LRR_1$  with the highest priority in H (i.e., between  $S_{0-2-3}$  and  $S_{0-4-3}$ ), (c-1) is  $LRR_3$  with the highest priority in H (i.e., between  $S_{0-2-3}$  and  $S_{0-4-3}$ ), (c-1) is  $LRR_3$  with the highest priority in H (i.e., between  $S_{0-2-3}$  and  $S_{0-4-3}$ ), (c-1) is  $LRR_3$  with the highest priority in H (i.e., between  $S_{0-3-3}$  and  $S_{0-4-3}$ ).

the highest priority in E (i.e., between  $S_{0-3-1}$  and  $S_{0-4-1}$ ), (c-2) is *LRR*2 with the highest priority in E (i.e., between  $S_{0-3-2}$  and  $S_{0-4-2}$ ).

Figure 8. the differences of indexes (i.e., *LRR*1, *LRR*2, *LRR*3 for log response ratio of the S, H, and E component) with IWDPs (i.e., between  $S_{3-p-n}$  and  $S_{3-4-n}$ ) at the monthly scale: (a-1) is *LRR*2 with the highest priority in S (i.e., between  $S_{3-1-2}$  and  $S_{3-4-2}$ ), (a-2) is *LRR*3 with the highest priority in S (i.e., between  $S_{3-1-3}$  and  $S_{3-4-3}$ ), (b-1) is *LRR*1 with the highest priority in H (i.e., between  $S_{3-2-1}$  and  $S_{3-4-1}$ ), (b-2) is *LRR*3 with the highest priority in H (i.e., between  $S_{3-2-3}$  and  $S_{3-4-3}$ ), (c-1) is *LRR*1 with the highest priority in E (i.e., between  $S_{3-3-1}$  and  $S_{3-4-1}$ ), (c-2) is *LRR*2 with the highest priority in E (i.e., between  $S_{3-3-2}$  and  $S_{3-4-3}$ ).

---

## Author Response (AR2)

**Cover letter**

April 21st, 2025

**Number:** hess-2024-399

**Manuscript Title:** Impacts of Inter-basin Water Diversion Projects on the Feedback Loops of Water Supply-Hydropower Generation-Environment Conservation Nexus

Dear Prof. Pieter van der Zaag,

We sincerely appreciate the time and effort you have dedicated to evaluating our manuscript titled "Impacts of Inter-basin Water Diversion Projects on the Feedback Loops of Water Supply-Hydropower Generation-Environment Conservation Nexus". We much appreciate your professional and insightful comments. All the concerns raised have been carefully treated and an itemized reply to your comments is presented in the revision files. Our changes are marked in Red in the revised manuscript.

Thank you very much again for your time and kind help. Looking forward to hearing from you.

Sincerely yours,

Dedi Liu
Email: dediliu@whu.edu.cn

**Number: hess-2024-399**

**RESPONSES TO EDITOR'S COMMENTS**

*COMMENT:* Thanks to the improvements made, the manuscript now reads much better, although it is still hard work to read the paper, and there remain many minor grammar weaknesses, which I do not have time to address. (But if the authors would share the word manuscript, I could make an attempt to edit it.)

*RESPONSE:* We sincerely appreciate your constructive feedback and the time you have dedicated to improving our manuscript. We are glad to hear that the revisions have enhanced readability and are grateful for your continued support. We fully understand the challenges posed by remaining grammatical inconsistencies and deeply value your offer to assist with further edits. As requested, we can provide a Word version of the manuscript. Your expertise in refining these details would be invaluable, and we are eager to incorporate your suggestions to elevate the manuscript's clarity and precision.

Please let us know if there are additional adjustments or sections you believe warrant priority attention. We remain committed to addressing all concerns thoroughly and are truly thankful for your guidance throughout this process.

**Point #1**

*COMMENT:* *At several instances the paper refers to "reservoirs group" (e.g. lines 15, 47, 84, 205, 207, 215, 217 etc.), which nowhere is defined. This definition is needed because "reservoirs group" is not an established concept.*

*RESPONSE:* We sincerely appreciate the editor's careful reading and constructive feedback. We agree that the term "reservoirs group" requires clearer definition given its non-standardized usage in existing literature. In this study, "reservoirs group" refers to multiple reservoirs operated in series and parallel configurations to collaboratively manage water resource development and utilization across the basin. In the revised manuscript, we will add the following definition in the Introduction section (Section 1) where the concept is first introduced (Line 47):

"From the perspective of reservoir nodes under scrutiny, current research primarily focuses on single reservoirs (Wu et al., 2021), virtual reservoirs (Chen et al., 2020), and cases of two connected reservoirs (Khalkhali et al., 2018). To optimize the allocation of basin-scale water resources, the deployment of cascade reservoir systems has increased significantly (Liu et al., 2022), wherein multiple reservoirs with different priority functions are strategically interconnected through series-parallel hydraulic linkages. These reservoirs establish a reservoirs group to collaboratively manage the basin's water resource development and utilization. However, few of them focus on the reservoirs group with different priority functions."

**Point #2**

**COMMENT:** *At several instances the paper uses the term "collaborative states" (lines 14, 85, 101, 597, 615). Is this the same as "synergies"? I am not sure whether "collaborative state" is a felicitous term, as it may suggest that there is a "volition" of working together, but that's not what this is about. In practice there is either a synergy or a trade-off or an absence of interaction. It is in my view noteworthy that in a nexus paper the word "synergy" is nowhere mentioned. Similarly, equally surprising is that the concept of trade-off is nowhere mentioned in the text.*

**RESPONSE:** We sincerely appreciate the editor's constructive feedback regarding the readability of the manuscript. The points raised are indeed crucial for enhancing the clarity and academic rigor of our nexus analysis. We fully agree with the reviewer that the term "collaborative states" could inadvertently imply a volitional or intentional dimension of interaction, which does not align with the objective nature of nexus. After careful consideration, we have replaced all instances of "collaborative states" with "synergies" throughout the manuscript (e.g., Lines 14, 87, 104, 601, 619) to better reflect the systemic interdependencies inherent in SHE systems.

**Point #3**

**COMMENT:** *Why not call NEXUS I -> NEXUS SH; NEXUS II -> NEXUS SE, and NEXIS III -> Nexus HE (see e.g. Fig 1, line 223, 258, 259, 260, 261, 262, 263, 265). Reads much more easily, as it is more intuitive.*

**RESPONSE:**
We sincerely thank the editor for raising this important point regarding the nomenclature of our

nexus scenarios. We deeply appreciate your effort to enhance the readability of our framework and acknowledge that our original labeling (Nexus I/II/III) may not have been fully explained. To clarify, the Nexus devices are defined as:

- **Nexus I**: *Nexus with inter-basin water diversion projects*.
- **Nexus II**: *Nexus without inter-basin water diversion projects*.
- **Nexus III**: *Nexus with the different clusters of inter-basin water diversion projects*.

While we agree that acronyms like SH/SE/HE could offer intuitive cues, since the representation of NEXUS is not easy to express in an abbreviated way, it may make the article more complicated. Therefore, we opted for Roman numerals to:

To address your concern, we have:

- Added an explanatory footnote in Fig. 1, explicitly defining the incremental logic of Nexus I-III.
- Added explanations of the Nexus I-III in Section 2.1 to help readers better understand their meanings.
- Added explanations of Nexus I-III in Table S6 in the Supplementary Material.

We sincerely regret any ambiguity caused by our original phrasing and are grateful for your insightful suggestion, which prompted us to significantly improve the transparency of our scenario definitions. Should the editor still consider acronyms preferable, we would be happy to adopt other way of naming.

"To address the impacts of IWDPs across the multiple temporal and spatial scales on the dynamic SHE nexus, multiple temporal and spatial scales runoffs from the water donating basins are provided through a distributed hydrological model. And multi-level ecological flows and their corresponding multi-level ecological flow standards are also determined according to an available method with spatial-temporal variability. To facilitate the identification of the impacts of IWDPs on SHE nexus, scenario experiments are set by "with/without IWDPs". In order to take the different clusters of IWDPs into account, scenario experiments are classified by the impacts of IWDPs on water donation area, on water receiving area or on an area with both water donation and water receiving if there are IWDPs. To evaluate the feedback loops of the SHE nexus, the priority order of S, H, and E are iteratively set in all reservoir nodes. We set different types of the highest priority in S, H, and E and take the standard scheduling rules as reference scenarios. All scenarios are modeled in a multisource input-output reservoir generalization model, and differences between scenarios are quantified with a response ratio indicator. And the feedback loops with the different impacts of IWDPs are identified through a response ratio indicator. To explore the synergies, positive mutation in a response ratio across time-space is found between pairwise components of SHE. This framework can be applied globally to identify the feedbacks of the SHE nexus in basins with IWDPs.

Thus, our research framework is illustrated as Figure 1. The Nexus I-III in Figure 1 are defined as the nexus with IWDPs,the nexus without IWDPs and the nexus with the different clusters of IWDPs.

[Figure]

**Figure 1. Framework to identify the impacts of different IWDPs on the feedback loops of SHE nexus.**"

"**Table S6**. List of Abbreviations

| Abbreviation | Full Term |
|---|---|
| IWDPs | Inter-basin water diversion projects |
| S | Water supply |
| H | Hydropower generation |
| E | Environment conservation |

| | |
|---|---|
| SHE | Water Supply-Hydropower Generation-Environment Conservation |
| Nexus I | Nexus with inter-basin water diversion projects. |
| Nexus II | Nexus without inter-basin water diversion projects. |
| Nexus III | Nexus with the different clusters of inter-basin water diversion projects. |
| HRB | Hanjiang River Basin |
| S-Priority | the highest priority is set to water supply |
| H-Priority | the highest priority is set to hydropower generation |
| E-Priority | the highest priority is set to environment conservation |
| the VIC model | The Variable Infiltration Capacity hydrological model |
| NSE | the Nash-Sutcliffe efficiency coefficient |
| $R^2$ | Coefficient of determination |
| PBIAS | Percent bias |
| the MTMMHC method | The Modified Tennant Method Based on Multilevel Habitat Conditions method |
| the MIORG model | The Multisource Input-Output Reservoir Generalization model |
| EFs | ecological flows |
| LRR | log response ratio |
| DEM | the Inverse Distance Weighting method. Digital Elevation Model |
| HWSD | the Harmonized World Soil Database |
| SWCT | the Soil-Water Characteristics |
| FAO | Food and Agriculture Organization |
| IIASA | Institute of Internal Auditors South Africa |

"

**Point #4**

*COMMENT: Table 3:*

*- Usable storage: why not use the more convention unit $10^6$ m³ rather than $10^8$ m³?*

*- "Annual generation" -> "Energy generation"*

*- Correct unit: billion kWh/yr (not billion kWh)*

*RESPONSE:* We sincerely appreciate the editor's constructive feedback on improving the clarity and comparability of our figures.

1) Unit of "Usable storage": Following your suggestion, we have converted the unit from $10^8$ m³

to $10^6\,\text{m}^3$ (e.g., "0.92" → "92") to adhere to widely recognized hydrological/metric standards.

2) Terminology Adjustment ("Annual generation" → "Energy generation"): The header has been revised to "Energy generation".

3) Unit Correction for Energy Generation: The unit is now explicitly stated as billion kWh/yr (instead of "billion kWh") to emphasize the annualized nature of the data.

"**Table 3**. List of characteristic parameter values of reservoirs.

| Characteristic parameter | Unit | Huang Jinxia | An Kang | Dan Jiangkou | Wang Fuzhou | Xing Long |
|---|---|---|---|---|---|---|
| Operational year | year | 2023 | 1992 | 2013 | 2003 | 2013 |
| Normal water level | m | 450 | 330 | 170 | 86.23 | 36.2 |
| Usable storage | $10^6\text{m}^3$ | 92 | 1680 | 16360 | 149.5 | 24.6 |
| Dead water level | m | 440 | 305 | 150 | 85.48 | 35.7 |
| Installed capacity | MW | 135 | 800 | 900 | 109 | 40 |
| Energy generation | billion kWh/yr | 0.25 | 2.80 | 3.83 | 0.58 | 0.23 |
| Comprehensive hydropower coefficient | kg/(s²·m²) | 8.4 | 8.4 | 7.7 | 8.5 | 8.4 |
| Regulation ability | time | Daily | Yearly | Multi-year | Daily | Daily |

"

**Point #5**

*COMMENT:*

*Typos / minor grammar issues in the manuscript:*

*a. Line 133: lager -> larger*

*b. Line 245: denoated -> denoted*

*c. Line 263: "can figure out" -> "show"*

*d. Line 279: "15 cascade reservoirs" -> "a cascade of 15 reservoirs"*

*e. Line 588: "in consisted with that in Hutuo River Basin" -> "consistent with those of the Hutuo River Basin" OR "inconsistent with those of the Hutuo River Basin" [I do not know which one of the two is meant. Carefully check!]*

*f. Lines 617-618: "will be figured out" -> "can be elaborated"*

*RESPONSE:* We sincerely apologize for the inadvertent linguistic oversights in our manuscript

and deeply appreciate your meticulous review, which has significantly improved the clarity and professionalism of our work. The corrections have been implemented as follows:

1) Line 137:

"lager" → "larger" (Corrected typographical error).

2) Line 249:

"denoated" → "denoted" (Revised spelling error).

3) Line 264 and 266:

"can figure out" → "show" (Adjusted informal phrasing to align with academic style).

4) Line 283:

"15 cascade reservoirs" → "a cascade of 15 reservoirs" (Revised for grammatical precision).

5) Line 592:

"in consisted with that in Hutuo River Basin" → "consistent with those of the Hutuo River Basin"

(We confirm the intended meaning was "consistent"; this revision clarifies the comparative analysis. We apologize for the ambiguous wording.)

6) Lines 621:

"will be figured out" → "can be elaborated" (Enhanced phrasing for technical rigor).

    Thank you once again for your invaluable support and dedication to scholarly rigor. We have checked and verified the word spelling and grammar of the entire article. We hope there will be no more minor mistakes.

---

## Author Response (AR3)

**Cover letter**

May 6th, 2025

**Number:** hess-2024-399

**Manuscript Title:** Impacts of Inter-basin Water Diversion Projects on the Feedback Loops of Water Supply-Hydropower Generation-Environment Conservation Nexus

Dear Prof. Pieter van der Zaag,

We sincerely appreciate the time and effort you have dedicated to evaluating our manuscript titled "Impacts of Inter-basin Water Diversion Projects on the Feedback Loops of Water Supply-Hydropower Generation-Environment Conservation Nexus". All the concerns raised have been carefully treated and an itemized reply to your comments is presented in the revision files. Our changes are marked in Red in the revised manuscript.

Thank you very much again for your kind help. Looking forward to hearing from you.

Sincerely yours,

Dedi Liu
Email: dediliu@whu.edu.cn

**RESPONSES TO EDITOR'S COMMENTS**

**Point #1**

*COMMENT:* I appreciate the improvements made. There are however still a few remaining issues that require clarification. One is the use of the term "collaboration" that I understand to mean, in the specific context of this paper, synergy.

*RESPONSE:* We deeply appreciate the editor's insightful critique regarding the semantic nuances between "collaboration" and "synergy", "collaborative loops" and "synergetic loops". We fully agree that our original terminology risked conflating intentional coordination with emergent convergence phenomena. As suggested, we have:

♦  Replaced all instances of " *collaboration* " with " *synergy* "(Lines 60 and 564)

♦  Replaced all instances of "*collaborative loops*" with "*synergetic loops*" (Lines 61)

**Point #2**

*COMMENT:* Another is the statements referring to the positive or negative impacts of feedbacks. Such formulations remain unclear, as what is a negative impact for one, may be a positive impact for another. So this requires a more neutral (I mean non-normative) description.

*RESPONSE:* We sincerely appreciate this insightful critique regarding the potential normative connotations of "positive/negative impacts" formulations. To achieve value-neutral characterization, we have implemented the following revisions:

"With IWDPs, the water donation basins experience strengthened feedback loops, while water receiving basins experience weakened feedbacks." (Lines 22-23)

"water donation strengthens the negative feedback of S on H and E for five reservoirs." (Line 447)

"IWDPs strengthen the negative feedbacks of S on H and E for HJX, AK, DJK and WFZ and weaken the negative feedbacks of S on E for XL." (Line 451-452)

"Thus, water donation is found to strengthen the feedbacks of H on S and E, especially in low flow months. If there was only water receiving and H-Priority was set, values of $LRR_1$ and $LRR_3$ for DJK, WFZ and XL are greater

than those without IWDPs as shown in Figure 11 (b-1) and (b-2). Water receiving weakens the feedbacks of H on S and E." (Line 460-462)

"Water donation strengthens feedbacks of H on S for HJX, DJK and XL." (Line 489)

"There are both negative and positive feedbacks of the E component on H while the negative feedbacks strengthen in abundant water months." (Line 542)

"It is evident that water donation strengthens the negative feedbacks between S and H, the negative feedbacks between S and E, and the positive feedbacks between H and E, while receiving water weakens these feedbacks. Water donation results in a reduction of available water (Mok et al., 2015; Wu et al., 2022), leads to lower flow, to stronger competition for water among S, H and E, and strengthen the feedbacks. Reduced competition among S, H and E is found in water receiving areas, primarily due to the replenishment of available water resources." (Lines 564-569)

**Point #3**

*COMMENT:* I also feel that one finding of your study, which I find the most important finding, is not featuring in the conclusion nor in the abstract. It is written in the discussion section as follows: "The persistent feedback polarity with IWDPs suggests that simply increasing water supply (e.g., via compensation donations like Three Gorges-to-Hanjiang) cannot resolve inherent SHE conflicts." This conclusion is important for me, because inter-basin transfer schemes are often portrayed to solve such water conflicts.

*RESPONSE:* We sincerely appreciate the editor's astute observation regarding the critical finding. To elevate its prominence:

⬧ Abstract Enhancement: Added in Line 25-27:

"Simply increasing water receiving cannot resolve inherent SHE conflicts because of the persistent feedback polarity with IWDPs. Adaptive allocation rules are needed that account for these stable feedback patterns."

⬧ Conclusion Restructuring: Added in Line 602-604:

"We find that simply increasing water receiving cannot resolve inherent SHE conflicts because of the persistent feedback polarity with IWDPs. Adaptive allocation rules are needed that account for these stable feedback patterns."

**Point #4**

*COMMENT:* Furthermore there are still many minor editorial issues, which I have indicated (in track changes) in the word file that I received from the authors, and which I will return to

them via email (as this copernicus environment doesn't allow me to attach word files, while a converted pdf would obscure certain editorial details). Therein I also make several suggestions for editorial improvements. I trust the authors will be able to address the above in a hopefully final round of revisions.

*RESPONSE:*

We extremely appreciate your meticulous attention to editorial details and the effort invested in annotating the manuscript via track changes. We confirm receipt of the annotated Word file via email and will rigorously address all editorial suggestions. Your granular feedback has been invaluable in elevating the manuscript's professionalism. Our changes are marked in Red in the track-changes file.

We wish to express our deepest gratitude for the exceptional professionalism, patience, and intellectual rigor you have demonstrated throughout the review process.